# BRIDGING THE DOMAIN GAP BY CLUSTERING-BASED IMAGE-TEXT GRAPH MATCHING

## ABSTRACT

Learning domain-invariant representations is important to train a model that can generalize well to unseen target task domains. Text descriptions inherently contain semantic structures of concepts and such auxiliary semantic cues can be used as effective pivot embedding for domain generalization problems. Here, we use multimodal graph representations, fusing images and text, to get domain-invariant pivot embeddings by considering the inherent semantic structure between local images and text descriptors. Specifically, we aim to learn domain-invariant features by (i) representing the image and text descriptions with graphs, and by (ii) clustering and matching the graph-based image node features into textual graphs simultaneously. We experiment with large-scale public datasets, such as CUB-DG and DomainBed, and our model achieves matched or better state-of-the-art performance on these datasets. Our code will be publicly available upon publication.

## 1    INTRODUCTION

How can humans effectively comprehend visual concepts despite variations in backgrounds, textures, and artistic styles? If it is impossible to collect sufficient examples of various combinations of domains, can current machine learning methods found on the i.i.d. assumption achieve robust generalization performance across domains? In this paper, we consider the domain generalization problem on image datasets and introduce a new novel graph neural network method to tackle the problem.

Domain generalization aims to improve a model's generalization ability for unseen task domains. To tackle this problem, many types of research have been recently suggested, such as reducing domain discrepancies in the visual feature space (Li et al., 2018b; Sun & Saenko, 2016; Ganin et al., 2016; Kim et al., 2021), augmenting data to cover various domains (Yan et al., 2020; Xu et al., 2021; 2020; Kang et al., 2022), and utilizing ensemble learning (Cha et al., 2021; Lee et al., 2022). Recently, a new approach (Min et al., 2022) utilizing auxiliary semantic cues, such as text descriptions, to get domain-invariant features is suggested.

In this paper, we suggest using multimodal graph representations to get effective domain-invariant pivot embeddings for domain generalization problems. From the previous work (Min et al., 2022), it is well known that the text information contains general descriptions across various image domains. Moreover, in this paper, we argue that the text descriptions inherently contain semantic structures of concepts. Therefore, it is crucial to consider the inherent semantic structure between local images and text descriptors to get effective pivot embedding. To implement this idea, we suggest representing the text descriptions with graphs and aligning the embedding of images and text embedding by matching the graphs.

The suggested method consists of three parts: a graph-based text encoder, a graph-based image encoder, and a clustering-based alignment between two graphs. To tackle the problem, each input is represented with graphs. Based on the graph-based representations, we aim to learn the domain-invariant features by grounding the graph-based image features into textual graphs, as the textual graphs contain explicitly verbalized knowledge from humans' typical reasoning. To solve the language grounding with structural information, we suggest a new method that clusters and matches the graph node features simultaneously. By matching the multimodal graphs while clustering each node's features, our suggested method can get robust domain-invariant features representing multi-level semantic alignment.

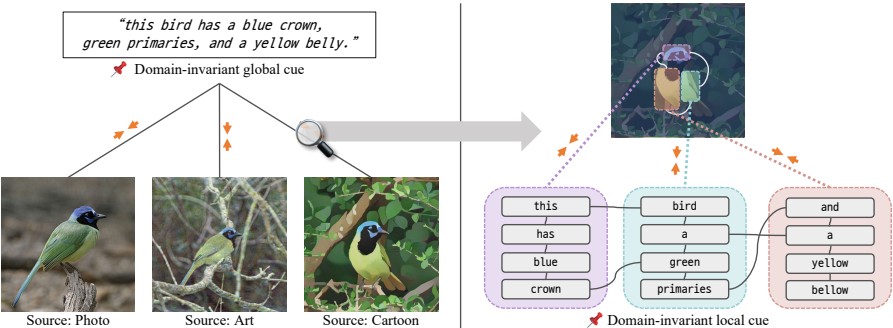

Figure 1: Our model learns to extract domain-invariant visual representations well-aligned with textual descriptions, e.g. "the bird has a blue crown, green primaries, and a yellow belly." Visual and textual features are modeled as a clustering-based graph such that our model learns to match a pair of visual and textual clusters, yielding domain-invariant visual features for models to generalize well in unseen target domains.

Experimental results and visualizations with two popular benchmark datasets, CUB-DG (Min et al., 2022) and DomainBed (Gulrajani & Lopez-Paz, 2020), show the pivotal role of multimodal structural representations. Quantitatively, the suggested method achieves a new state-of-the-art performance, especially by increasing generalization ability on the most difficult domain *paint*. With robust qualitative visualization results, we argue that our model learns domain-invariant features across various feature resolutions by locally and globally aligning with textual graphs. Our contributions can be summarized as follows.

- We suggest the first approach using graph representations for both image and text inputs for the domain generalization problem.
- We suggest a new graph neural network method that clusters and matches node features concurrently to align given two multimodal graphs.
- We achieve a new state-of-the-art domain generalization performance on the CUB-DG dataset and Domainbed benchmark.

## 2 RELATED WORK

**Domain Generalization.** Domain generalization refers to the task of improving a model's generalization performance on unseen target domains where data distribution differs from the source domains. The main idea of domain generalization is to learn domain-invariant features from multiple source domains. Various methods have been proposed to resolve this problem by (i) reducing domain discrepancies in the feature space (Li et al., 2018b; Sun & Saenko, 2016; Ganin et al., 2016; Kim et al., 2021), (ii) by implementing data augmentation (Yan et al., 2020; Xu et al., 2021; 2020; Kang et al., 2022), and (iii) by utilizing ensemble learning (Cha et al., 2021; Lee et al., 2022). (iv) Other studies have proposed using auxiliary semantic cues to facilitate learning domain-invariant features (Bai et al., 2021; Shahtalebi et al.; Cha et al., 2022). For instance, DecAug (Bai et al., 2021) uses context information, such as backgrounds, to disentangle spurious correlations that can cause domain shift problems.

Recently, GVRT (Min et al., 2022) successfully leverages textual descriptions for models to learn domain-invariant visual representations by aligning them with verbalized (domain-invariant and class-discriminative) knowledge from humans' typical reasoning (e.g., given a text "this bird is black with an orange spot on its wing"). Following the similar line of GVRT (Min et al., 2022), we also want to improve the model's generalization power by leveraging visual and textual inputs together. However, we focus more on aligning locally-aware high-order semantic relations via graph structures instead of simply matching global representations.

**Graph Neural Network.** Along with the huge success of neural networks in computer vision and natural language processing domains, new methodologies to deal with irregular structural inputs have been recently suggested. To learn the representations from the structural inputs, such as molecular graphs, social networks, and meshes, various types of graph-based neural network

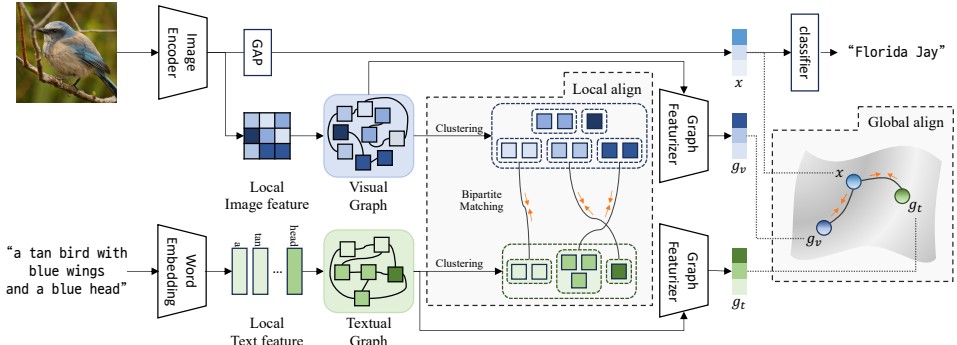

Figure 2: An overview of our proposed method. Our model consists of three main modules: (i) a Graph-based Visual Encoder, (ii) a Graph-based Textual Encoder, and (iii) Local and Global Alignment between Visual and Textual Graphs. Local image features (from the backbone) are represented as a graph structure (see Visual Graph), while word embeddings from textual descriptions are also represented as another graph structure (see Textual Graph). These multimodal graphs are then aligned with each other locally and globally, yielding domain-invariant visual features that are well-aligned with humans' explicitly verbalized knowledge.

algorithms are suggested. According to the ways of representing graph data, convolution-based methods(ChebyNet (Defferrard et al., 2016), GCN (Kipf & Welling, 2016)), attention-based methods(MoNet (Burgess et al., 2019), GAT (Veličković et al., 2017)), and message-passing methods(MPNN (Gilmer et al., 2017), GraphNets (Battaglia et al., 2018)) can be applied to graph representation learning. Those graph neural network methods have achieved great performance on graph-related tasks, such as node classification (Bhagat et al., 2011), link prediction (Zhang & Chen, 2018), and graph classification problems (Zhang et al., 2018), by leveraging the non-euclidean data manifolds to get informative representations. Recently, the applications of graph neural networks have been extended to image and text domains (Liu et al., 2020). By representing the image and text inputs as graphs, it becomes possible to consider the irregular and high-order correlations between tokens. In this paper, we suggest representing the multimodal inputs as graphs and matching the semantic correspondences between the multimodal inputs using graph neural networks to get the domain-invariant features.

## 3 METHOD

Given a distribution over multiple (or single) source domains $\{\mathcal{S}_1, \mathcal{S}_2, \dots\} \in \mathcal{S}$, the domain generalization (DG) problem considers the following classical stochastic optimization, in which we minimize the data-dependent generalization upper bound of the expected task loss (Sinha et al., 2017):

$$\underset{\theta}{\text{minimize}} \sup_{\mathcal{T}:\mathcal{D}(\mathcal{S},\mathcal{T})\leq\rho} \mathbb{E}_{\mathcal{T}}\big[\mathcal{L}(\theta;\mathcal{S})\big] \tag{1}$$

where we consider unseen target domains $\mathcal{T} = \{\mathcal{T}_1, \mathcal{T}_2, \dots\}$ and the discrepancy between $\mathcal{S}$ and $\mathcal{T}$ is bounded by an arbitrary bound $\rho$, i.e. $D(\mathcal{S},\mathcal{T}) \leq \rho$. Here, we consider image classification scenarios and we define the task-specific loss $\mathcal{L}$ function by the cross-entropy loss.

Extracting domain-invariant representations from an input image is substantially important to train such a model that generalizes well to unseen target domains. Inspired by recent work by Min et al. (2022), we also want to improve the model's generalization power by leveraging visual and textual inputs together – i.e. our model learns to extract (domain-invariant) visual representations that are well-aligned with explicitly verbalized knowledge from humans' typical reasoning. Our model is, however, different from Min et al. (2022) in that we focus more on aligning locally-aware high-order semantic relations via graphs instead of simply matching global representations.

Our model consists of three main parts: (i) a Graph-based Visual Encoder, (ii) a Graph-based Textual Encoder, and (iii) Local and Global Alignment between Visual and Textual Graphs. In (i), local latent representations (from a backbone network) are represented as a graph structure. Each local latent vector becomes a node, creating edges based on pairwise similarity in the embedding space

(see Section 3.1). In (ii), we build a textual graph given a natural language description about a specific class (e.g., "Ruby Throated Hummingbird has a long pointed bill with a white belly and a black crown."). Each word embedding becomes a node, similarly creating edges based on embedding-level similarity (see Section 3.2). Lastly, in (iii), we regularize multi-modal encoders to produce locally aligned representations by minimizing graph-level distances between visual and textual representations such that our model learns human-compatible visual cues, thus, becomes more generalizable (see Section 3.3).

## 3.1 GRAPH-BASED VISUAL ENCODER

**Global Visual Feature Extraction.** Following standards in the domain generalization task, we use the same backbone ResNet50 (He et al., 2016b) to take images $\mathcal{I}$ as an input, outputting a $d$-dimensional global visual representation $\mathbf{x}_g \in \mathbb{R}^d$. This global representation $\mathbf{x}_g$ is trained to predict its classification label $\hat{\mathbf{y}}$ with a linear layer, yielding the per-class softmax probabilities. Our backbone and a classifier are trained by a classification loss $\mathcal{L}_c$ as follows:

$$\mathcal{L}_c(\mathbf{y}, \hat{\mathbf{y}}) = - \sum_i y_i \log(\hat{y}_i) \tag{2}$$

where $\mathbf{y} \in \mathbb{R}^{|C|}$ is the ground-truth one-hot vector for all classes. Similar to findings in the literature, we also observe that minimizing the loss function $\mathcal{L}_c$ often leads the model to become semantically shallow: i.e. network's reasoning is prone to be simplified to a situation-specific dependence on salient visual cues (e.g. a black crown to classify Ruby-throated Hummingbird). These models would not generalize well in environments different from those in which they were trained. In our work, we aim to regularize our model to understand relations between visual cues and use those relations for the final verdict, thus making it more generalizable. We want to achieve such a regularization effect through utterances from human verbalized reasoning.

**Locally-aware Visual Graph Construction.** We first construct a graph with visual representations to achieve the abovementioned goal. Formally, given $M$ number of $d$-dimensional local visual representations $\mathbf{x}_l \in \{\mathbf{x}_{l,1}, \mathbf{x}_{l,2}, \ldots, \mathbf{x}_{l,M}\}$ extracted from intermediate layers of the backbone (before global average pooling layer) we consider these representations as a set of unordered nodes. Note that each representation vector $\mathbf{x}_{l,i} \in \mathbb{R}^d$ for $i \in [1, M]$ corresponds to a certain grid over an input image $\mathcal{I}$, as shown in Figure 2 (see Visual Graph). Inspired by the recent work by Han et al. (2022), we construct a graph such that each node $\mathbf{x}_{l,i}$ has an edge with the other $K_v$ nearest neighbors. We use the widely-used $L_2$ distance to measure pairwise node similarity. In summary, our visual graph $\mathcal{G}_v = (\mathcal{V}_v, \mathcal{E}_v)$ consists of $M$ nodes (representing each visual feature of the local grid) where each node is connected to its $K_v$ nearest neighbors.

**Graph-based Visual Representation.** Given the visual graph $\mathcal{G}_v$, we further apply two layers of graph convolution network (GCN) (Kipf & Welling, 2016) followed by a linear and a Batch-Norm (Ioffe & Szegedy, 2015) layers to learn relational knowledge between local visual representations (e.g. a bird who has a white belly, a white breast, a white, and a long hooked bill together is Laysan Albatross). Formally, we use a GCN-based function $f_{\text{GCN}}(\mathcal{G}_v)$ to obtain a final $d_g$-dimensional locally-aware visual graph representation $\mathbf{g}_v \in \mathbb{R}^{d_g}$: $\mathbf{g}_v = \frac{1}{M} \sum_i^M \mathcal{V}'_{v,i}$, where $\mathcal{G}'_v = (\mathcal{V}'_v, \mathcal{E}'_v)$ is obtained by $\mathcal{G}'_v = f_{\text{GCN}}(\mathcal{G}_v)$. Note that, we also add an additional classifier that takes the $\mathbf{g}_v$ as an input to create a graph that better captures the characteristics of the class. We provide detailed explanations in the Appendix.

## 3.2 GRAPH-BASED TEXTUAL ENCODER

**Word-level Textual Graph Construction.** Our graph-based visual representation $\mathbf{g}_v$ encodes relational knowledge via a graph structure between representations of local visual features $\mathbf{x}_l$. We empirically observe that such graph-based representation's sole use is still insufficient for models to learn domain-invariant and human-compatible visual cues. Thus, to regularize our visual encoders to be aligned with human knowledge, we build a textual graph from a natural language description of each class, followed by aligning both visual and textual graphs. A sequence of $L$ (at maximum) words is first tokenized and encoded with a standard word-level (learnable) embedding layer, producing $d_t$-dimensional embedding vectors $\mathbf{t} \in \{\mathbf{t}_1, \mathbf{t}_2, \ldots, \mathbf{t}_L\}$ where $\mathbf{t}_i \in \mathbb{R}^{d_t}$. Similar to our Visual Graph $\mathcal{G}_v$, we consider these word embeddings as an unordered set, and we construct a graph

such that each node $\mathbf{t}_i$ has an edge with the other $K_t$ nearest neighbors. We use $L_2$ distance to measure pairwise node similarity (see Figure 2). Finally, we create a textual graph $\mathcal{G}_t = (\mathcal{V}_t, \mathcal{E}_t)$.

**Graph-based Textual Representation.** Given the textual graph $\mathcal{G}_t$, we apply the same architecture (but not shared) to obtain textual graph representation $\mathbf{g}_t \in \mathbb{R}^{d_g}$. I.e. we apply another GCN-based function $f_{\text{GCN}}(\mathcal{G}_t)$ to learn relational knowledge between word embeddings: $\mathbf{g}_t = \frac{1}{L} \sum_i^L \mathcal{V}'_{t,i}$, where $\mathcal{G}'_t = (\mathcal{V}'_t, \mathcal{E}'_t)$ is obtained by $\mathcal{G}'_t = f_{\text{GCN}}(\mathcal{G}_t)$.

## 3.3 LOCAL AND GLOBAL ALIGNMENT BETWEEN VISUAL AND TEXTUAL GRAPHS

We apply the following two graph-matching approaches: (i) Global Graph Matching and (ii) Clustering-based Fine-grained Graph Matching.

**Global Graph Matching between $\mathbf{g}_v$ and $\mathbf{g}_t$.** A standard approach to matching two different graph representations is minimizing the Euclidean distance as follows:

$$\mathcal{L}_{\text{global}} = ||f_{\text{proj},x}(\mathbf{x}_g) - f_{\text{proj},v}(\mathbf{g}_v)||_2 + ||f_{\text{proj},x}(\mathbf{x}_g) - f_{\text{proj},t}(\mathbf{g}_t)||_2 \tag{3}$$

where we use a linear layer to project each feature (i.e. $\mathbf{x}_g$, $\mathbf{g}_v$, and $\mathbf{g}_t$) such that these three projected features are pulled together. Note that $f_{\text{proj},x}$, $f_{\text{proj},v}$, and $f_{\text{proj},t}$ represent a projection layer.

Importantly, as we use only a force to pull latent representations together, the training dynamics may become unstable, causing a representation collapse. We add an auxiliary classifier that takes the $f_{\text{proj},x}(\mathbf{x}_g)$ as an input to prevent such a mode collapse, outputting the per-class softmax probability. This classifier is trained with the standard cross-entropy loss.

**Clustering Graph Nodes.** In addition to global graph matching, we further apply Clustering-based Local Graph Matching, which aligns node-level features with regularizing two multi-modal graphs to use similar semantic cues (e.g. aligning visual features of a beak with a text embedding for "a long sharp beak"). However, simply aligning nodes from two different graphs may not work as these nodes have different representations (i.e. a visual feature of a local image region vs. a word-level representation). Thus, we apply a graph clustering algorithm such that two graphs have the same level of semantic representation. We define user-specified parameters $N_v (\leq M)$ and $N_t (\leq L)$ to the number of clusters for our visual and textual graphs, respectively. Note that we set $N_v \geq N_t$ since images may contain visual contents (e.g. backgrounds) that are not generally described in the text.

Our approach to constructing a graph is based on measuring node similarity, which can result in a well-defined semantic structure in the graph. Therefore, we choose a modularity-based method for graph clustering that can reflect this semantic structure while remaining stable. Specifically, we use a deep learning-based modularity measurement method (Tsitsulin et al., 2020). Our model first encodes the cluster assignment matrix using the features of the graph nodes. Then, we calculate the modularity using this matrix, which measures the quality of the clustering. We train the model to maximize the modularity while also constraining it with collapse regularization to prevent trivial solutions such as assigning all nodes to the same cluster. We formulate it as follows:

$$\mathcal{L}_d = -\frac{1}{2m} \text{Tr}(\mathbf{C}^{\text{T}}\mathbf{B}\mathbf{C}) + \frac{\sqrt{k}}{n} \left\| \sum_i \mathbf{C}_i^{\text{T}} \right\|_F - 1 \tag{4}$$

where $\mathbf{C}$ is the cluster assignment matrix calculated with our graph feature, and $\mathbf{B}$ is the modularity matrix calculated with the adjacency matrix. $m$, $n$, and $k$ represent the number of edges, the number of nodes, and the number of clusters, respectively. The first term refers to modularity, which is measured using the assignment matrix and the modularity matrix, while the second term represents the collapse regularization term. By doing so, our model can cluster semantically similar nodes together, allowing us to proceed with the matching process.

**Graph Cluster Matching.** Inspired by previous work (Carion et al., 2020), we use the set-based loss, i.e. the bipartite matching loss, between two disjoint sets of clusters: (i) a set of clusters $\mathcal{C}_v \in \{\mathcal{C}_v^1, \mathcal{C}_v^2, \ldots, \mathcal{C}_v^{N_v}\}$ of the visual graph $\mathcal{G}'_v$ and (ii) a set of clusters $\mathcal{C}_t \in \{\mathcal{C}_t^1, \mathcal{C}_t^2, \ldots, \mathcal{C}_t^{N_t}\}$ from the textual graph $\mathcal{G}'_t$. We minimize the following pair-wise matching loss: $\mathcal{L}_p = \frac{1}{N_t} \sum_{i=1}^{N_t} ||\mathcal{C}_v^{\mu_i} - \mathcal{C}_t^i||_2$, where $\mu_i \in \{1, 2, \ldots, N_v\}$ is the node index of the cluster in $\mathcal{C}_v$ which matches to $i$ in $\mathcal{C}_t$, producing the smallest total Euclidean distance by bipartite matching.

As the pair-wise matching loss pulls positive pairs together, negative pairs to add a repulsive force may need to prevent representation collapse. Thus, we also use a hinge loss based on that $\mathcal{C}_v^i$ and $\mathcal{C'}_t^j$ (where $i \in [1, N_v]$ and $j \in [1, N_t]$) forms a negative pair if they are clusters for different input images. Thus, the matched distance $L_p$ should be smaller than any other pairs between $\mathcal{C}_v^j$ and $\mathcal{C'}_t^i$ (or $\mathcal{C'}_v^j$ and $\mathcal{C}_t^i$). We formulate it as a hinge loss as follows:

$$\mathcal{L}_h = \max(0, \mathcal{L}_p - \text{MinDist}(\mathcal{C'}_v, \mathcal{C}_t) + \epsilon) + \max(0, \mathcal{L}_p - \text{MinDist}(\mathcal{C}_v, \mathcal{C'}_t) + \epsilon) \quad (5)$$

where $\text{MinDist}(\mathcal{C'}_v, \mathcal{C}_t)$ represents the minimum pair-wise matching loss similar to $\mathcal{L}_p$, but is applied to different inputs within a mini-batch. We compute it over all pairs of samples in a mini-batch and use the average as the final loss value: $\mathcal{L}_{\text{local}} = \frac{1}{B} \sum_b (\lambda_p \mathcal{L}_p + \lambda_h \mathcal{L}_h + \lambda_d \mathcal{L}_d + \lambda_{\text{aux}} \mathcal{L}_{\text{aux}})$, where we set the size of a mini-batch to $B$ and $\lambda_p$, $\lambda_h$, and $\lambda_d$ adjustable hyper-parameters that control the weight of each loss term. In our model, values of 0.1, 0.1, and 1 are used for $\lambda_p$, $\lambda_h$, and $\lambda_d$, respectively. Note that, similar to our global graph matching module, we also add an auxiliary classifier that takes the average-pooled cluster representation $\frac{1}{N_t} \sum_{i=1}^{N_t} \mathcal{C}_v^{\mu_i}$ as an input and outputs the per-class softmax probability, trained with the standard cross-entropy loss $\mathcal{L}_{\text{aux}}$.

**Loss Function.** Ultimately, we train our model end-to-end by minimizing the following loss $L$:

$$\mathcal{L} = \mathcal{L}_c + \lambda_{\text{global}} \mathcal{L}_{\text{global}} + \lambda_{\text{local}} \mathcal{L}_{\text{local}} \quad (6)$$

where $\lambda_{\text{global}}$, and $\lambda_{\text{local}}$ are hyper-parameters to control the strength of each loss term.

## 4 EXPERIMENTS

**Implementation Details.** Same as previous domain generalization approaches, we also use ImageNet (Deng et al., 2009) pre-trained ResNet-50 (He et al., 2016a) as our backbone, yielding a 2,048-dimensional visual representation from the last layer. Our model is trained end-to-end for 5,000 training steps using Adam optimizer with a learning rate of 5e-5. For training, we use standard image augmentation techniques such as random cropping, horizontal flipping, color jittering, grayscale conversion, and normalization. Our implementation is based on DomainBed (Gulrajani & Lopez-Paz, 2020), which is a unified domain generalization testbed, and our code will be publicly available upon publication. For hyperparameters, we use both 1 for $\lambda_{global}$ and $\lambda_{local}$, respectively. More details are available in Appendix.

**Datasets.** Our model learns domain-invariant features by representing the image and text descriptions with graphs, where their graph-based node features are clustered and matched. Therefore, our model is advantageous in using global and local information, learning subtle discriminatory features generally in fine-grained image classification tasks. To demonstrate its effectiveness, we first use the **CUB-DG** dataset (for fine-grained image classification task), which is extended from the CUB dataset (Welinder et al., 2010) for the domain generalization task. This dataset contains 11,768 images for 200 classes of North American bird species. Each image has up to 10 text descriptions describing the content in detail, e.g., "this bird is black with an orange spot on its wing." Each image is manipulated for the domain generalization task to create the following four domains: Photo, Cartoon, Art, and Painting. We follow the common evaluation protocol and use the CUB-DG dataset's official split (the train and validation set has 5,994 samples, while the test set has 5,794 samples).

Further, we also evaluate our model on DomainBed (Gulrajani & Lopez-Paz, 2020), which contains the following five multi-domain DG datasets: VLCS (Fang et al., 2013), PACS (Li et al., 2017), OfficeHome (Venkateswara et al., 2017), TerraIncognita (Beery et al., 2018), and DomainNet (Leventidis et al., 2021). Among these, we would emphasize that **PACS** (Li et al., 2017) dataset is useful for our experiments as (i) it provides a bigger domain shift than existing photo-only benchmarks, and (ii) it needs to exploit local information to learn discriminative subtle visual features. Note also that **TerraIncognita** (Beery et al., 2018) contains small ROI objects captured in the blurred or low illumination environment, which are difficult to spot even for humans. This would not be a good benchmark for our setting as we focus on learning objects' fine-grained features by aligning with textual descriptions. However, we still report scores on this dataset. We follow the standard evaluation protocol used in Gulrajani & Lopez-Paz (2020). For datasets that do not provide text inputs, we use both (1) textual class definitions from the Oxford dictionary similar to Min et al. (2022) and (2) descriptions generated by InstructBLIP (Dai et al., 2023) with the prompt "write a detailed description about the image." (see Appendix C).

Table 1: The out-of-distribution classification accuracies (in %) on CUB-DG (top) and PACS (bottom) datasets based on the standard leave-one-out multi-source DG task setting (i.e., a single domain is used as a target domain, while others are used as source domains). We compare ours with other existing DG approaches, but we only provide top-5 methods due to space constraints (we provide full tables in Appendix). *Abbr.* I: Image, T: Text.

| Algorithms (CUB-DG (Min et al., 2022)) | Modality | Target Domain | | | | Avg. ↑ |
|---|---|---|---|---|---|---|
| | | Photo | Cartoon | Art | Paint | |
| MIRO (Cha et al., 2022) | I | 68.2 | 59.1 | 46.5 | 38.2 | 53.0 |
| SD (Pezeshki et al., 2020) | I | 71.3 | 62.2 | 50.8 | 34.8 | 54.7 |
| CORAL (Sun & Saenko, 2016) | I | 72.2 | 63.5 | 50.3 | 35.8 | 55.4 |
| GVRT (Min et al., 2022) | I+T | 74.6 | 64.2 | 52.2 | 37.0 | 57.0 |
| Ours | I+T | **75.4** (0.8%↑) | **65.5** (1.4%↑) | **54.0** (1.8%↑) | **41.4** (4.4%↑) | **59.1** (2.1%↑) |

| Algorithms (PACS (Li et al., 2017)) | Modality | Art Painting | Cartoon | Photo | Sketch | Avg. ↑ |
|---|---|---|---|---|---|---|
| SelfReg (Kim et al., 2021) | I | 87.9 ± 1.0 | 79.4 ± 1.4 | 96.8 ± 0.7 | 78.3 ± 1.2 | 85.6 |
| CORAL (Sun & Saenko, 2016) | I | 88.3 ± 0.2 | 80.0 ± 0.5 | 97.5 ± 0.3 | 78.8 ± 1.3 | 86.2 |
| mDSDI (Bui et al., 2021) | I | 87.7 ± 0.4 | 80.4 ± 0.7 | 98.1 ± 0.3 | 78.4 ± 1.2 | 86.2 |
| SagNet (Nam et al., 2021) | I | 87.4 ± 1.0 | 80.7 ± 0.6 | 97.1 ± 0.1 | 80.0 ± 0.4 | 86.3 |
| Ours | I+T | **87.9 ± 0.7** | **81.4 ± 0.1** | **98.0 ± 0.1** | **80.5 ± 1.1** | **87.0** (0.7%↑) |

**Performance Comparison with Other DG Approaches.** As shown in Table 1, we start by comparing the out-of-distribution classification accuracies on the following two datasets: CUB-DG (top) and PACS (bottom) datasets. We compare ours with other existing domain generalization approaches, including SagNet (Nam et al., 2021), MIRO (Cha et al., 2022), SD (Pezeshki et al., 2020), CORAL (Sun & Saenko, 2016), GVRT (Min et al., 2022), SelfReg (Kim et al., 2021), and mDSDI (Bui et al., 2021). Due to space constraints, we only report top-5 results (we provide full tables in Appendix.).

As shown in Table 1 (top), our proposed method clearly outperforms the other domain generalization techniques on the CUB-DG dataset in all target domains with a significant gain. In terms of the average image classification accuracy, ours shows 59.1%, which is 2.1% higher than GVRT (Min et al., 2022) (which uses the same image and text inputs) and 3.7%-14.3% higher than other image-only approaches. Similar trends are also observed in our experiment on the large-scale PACS (Li et al., 2017) dataset. As shown in Table 1 (bottom), our model also outperforms the other approaches, i.e., ours shows 87.0% that is 0.7% higher than SagNet (Nam et al., 2021)-based SOTA approach and 1.9% higher than GVRT (Min et al., 2022). These confirm that our graph-based approach is effective in aligning visual and textual encoders for fine-grained image classification tasks, improving the visual encoder's generalization power to unseen target domains.

**Few-shot DG Performance Comparison.** Conventional DG approaches often assume that a sufficient number of images is available for all classes and domains enough to learn domain-invariant class-discriminative features. However, this may be practically challenging in real-world scenarios. We emphasize that our method, which leverages textual descriptions as pivotal information, can benefit learning domain-invariant features in the few-shot setting. Our model significantly outperforms the other approaches on PACS (Li et al., 2017) and VLCS (Fang et al., 2013) datasets. Note that we use randomly chosen five images (per class in each domain) as an input to train all models (i.e., 5-shot DG). Except for the number of training images, we generally follow the standard protocol of DomainBed for evaluation.

Table 2: Few-shot DG performance comparison. We report 5-shot (i.e., models are trained only with randomly chosen five images per class and per domain) DG performance compared with other SOTA approaches on the following two datasets: PACS (left) and VLCS (right).

| Algorithm (Data: PACS) | Avg. | Algorithm (Data: VLCS) | Avg. |
|---|---|---|---|
| mDSDI (Bui et al., 2021) | 63.5 | mDSDI (Bui et al., 2021) | 68.5 |
| CORAL (Sun & Saenko, 2016) | 64.6 | GVRT (Min et al., 2022) | 69.4 |
| MIRO (Cha et al., 2022) | 65.5 | MIRO (Cha et al., 2022) | 69.6 |
| GVRT (Min et al., 2022) | 68.7 | CORAL (Sun & Saenko, 2016) | 71.1 |
| Ours | **70.7** | Ours | **71.4** |

**Analysis of Graph Clusters and Their Matchings.** In Figure 3, we provide examples of a matched pair of image regions and a set of words. For example, in (a), a region around the bird's head is matched with a textual graph cluster that contains words including "red crown" and "cheek patch."

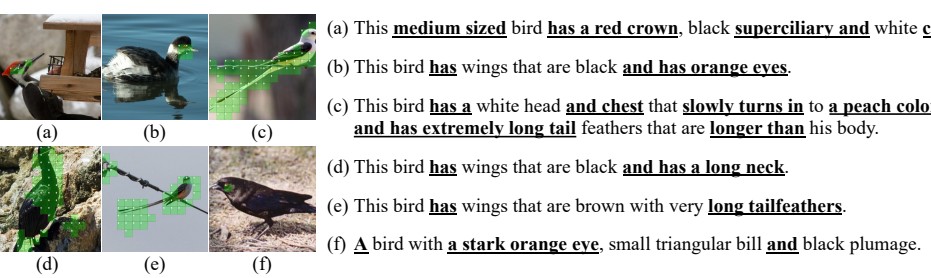

(a) This **medium sized** bird **has a red crown**, black **superciliary and** white **cheek patch**.

(b) This bird **has** wings that are black **and has orange eyes**.

(c) This bird **has a** white head **and chest** that **slowly turns in** to **a peach color near his feet, and has extremely long tail** feathers that are **longer than** his body.

(d) This bird **has** wings that are black **and has a long neck**.

(e) This bird **has** wings that are brown with very **long tailfeathers**.

(f) **A** bird with **a stark orange eye**, small triangular bill **and** black plumage.

Figure 3: Examples of the matched image region (in visual graph clusters) and texts (in textual graph clusters).

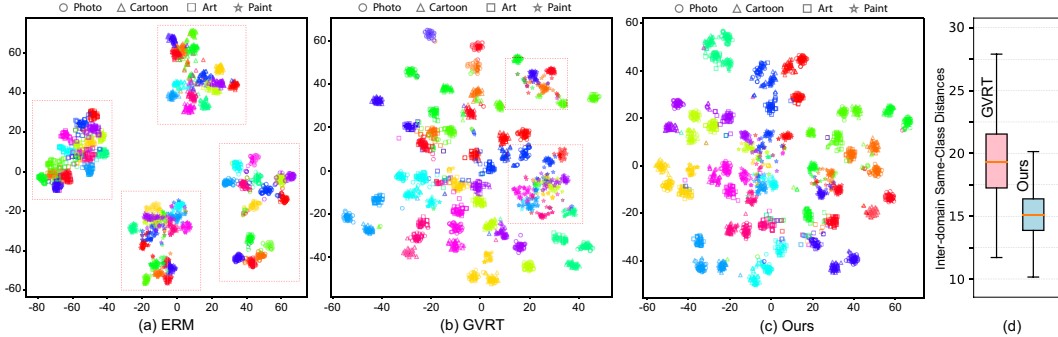

Figure 4: Visualizations by t-SNE for (a) ERM (Vapnik, 1999), (b) GVRT (Min et al., 2022), and (c) Ours on CUB-DG dataset. Each point is color-coded differently according to its class and has different shapes according to its domain. (d) We also compare inter-domain same-class distances.

We observe that our model reasonably matches image features with class-discriminative texts, e.g., orange eye, extremely long tail, and long neck.

**Bipartite vs. Greedy Matching.** Our model uses bipartite matching, a one-to-one matching process between image and text clusters, minimizing the distance between the matched cluster pairs. In Table 3, we compare it with greedy matching, which associates each text cluster with the nearest image cluster based on spatial proximity (i.e., allowing many-to-one correspondence). We observe that ours with bipartite matching generally outperforms greedy matching, possibly due to the constraints for 1-to-1 matching yielding a dispersive effect for models to optimize cluster pairs globally.

Table 3: Performance comparison between variants of our model with different matching techniques: bipartite matching and greedy matching. Data: CUB-DG.

| Bipartite Matching | Greedy Matching | Target Domain | | | | Avg. |
|---|---|---|---|---|---|---|
| | | Photo | Cartoon | Art | Paint | |
| - | ✓ | 75.3 | 64.7 | **54.8** | 36.6 | 57.8 |
| ✓ | - | **75.4** | **65.5** | 54.0 | **41.5** | **59.1** |

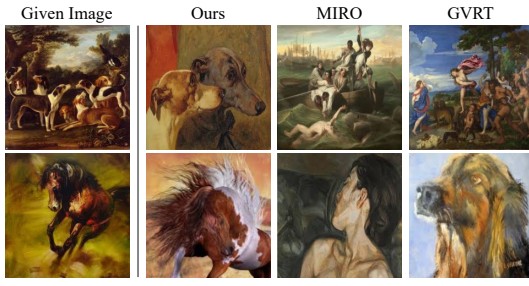

Figure 5: Exemplars of the nearest examples from PACS dataset (in the unseen target domain) to the given image (e.g., "dog").

**t-SNE Analysis.** As shown in Figure 4, we provide t-SNE (Van der Maaten & Hinton, 2008) visualization of (a) ERM (Vapnik, 1999), (b) GVRT (Min et al., 2022), and (c) ours to visualize their embedding space on CUB-DG dataset. We use different marker styles (for target domains) and different colors (for classes). An ideal model would show that visual features of the same class but different domains are gathered together. GVRT and ours clearly outperform ERM, which has scattered points per domain. Also, ours is better than GVRT in that features of the same class but different domains are more clustered (see red boxes). In Figure 4 (d), we provide box plots for GVRT and ours, showing that our model produces lower same-class inter-domain distances than GVRT. Note that we provide detailed t-SNE visualizations in Appendix.

Table 4: Ablation studies to compare variants of our model with and without global graph alignment (i.e. graph-level feature matching) and (ii) local graph alignment (i.e. clustering-based graph node matching), (iii) Visual Graph, and (iv) Textual Graph. Data: CUB-DG.

| Global Alignment $\mathcal{L}_{global}$ | Local Alignment $\mathcal{L}_{local}$ | Visual Graph $\mathcal{G}_v$ | Textual Graph $\mathcal{G}_t$ | Target Domain | | | | Avg. |
|---|---|---|---|---|---|---|---|---|
| | | | | Photo | Cartoon | Art | Paint | |
| - | - | ✓ | ✓ | 65.1 | 52.5 | 38.2 | 29.0 | 46.2 (12.9%↓) |
| - | ✓ | ✓ | ✓ | 71.4 | 61.3 | 49.4 | 34.5 | 57.2 (1.9%↓) |
| ✓ | ✓ | ✓ | ✓ | **75.4** | **65.5** | **54.0** | **41.5** | **59.1** |
| ✓ | ✓ | - | ✓ | 75.0 | 64.4 | 53.0 | 34.7 | 56.8 (2.3%↓) |
| ✓ | ✓ | ✓ | - | 70.3 | 57.0 | 48.1 | 33.5 | 52.2 (6.9%↓) |
| ✓ | ✓ | - | - | 68.5 | 59.0 | 38.6 | 32.5 | 49.6 (9.5%↓) |

Table 5: The test accuracies (in %) on the DomainBed datasets in the multi-source DG task setting. We compare ours with the existing 19 other DG approaches (we provide top-5 results here, and full table is available in Appendix). *Abbr.* I: Image, T: Text.

| Algorithm | Modality | Dataset | | | | | Avg. |
|---|---|---|---|---|---|---|---|
| | | VLCS | PACS | OfficeHome | TerraIncognita | DomainNet | |
| SagNet (Nam et al., 2021) | I | 77.8 ± 0.5 | 86.3 ± 0.2 | 68.1 ± 0.1 | 48.6 ± 1.0 | 40.3 ± 0.1 | 64.2 |
| CORAL (Sun & Saenko, 2016) | I | 78.8 ± 0.6 | 86.2 ± 0.3 | 68.7 ± 0.3 | 47.6 ± 1.0 | 41.5 ± 0.1 | 64.6 |
| mDSDI (Bui et al., 2021) | I | 79.0 ± 0.3 | 86.2 ± 0.2 | 69.2 ± 0.4 | 48.1 ± 1.4 | 42.8 ± 0.1 | 65.1 |
| GVRT (PTE) (Min et al., 2022) | I+T | 79.0 ± 0.2 | 85.1 ± 0.3 | 70.1 ± 0.1 | 48.0 ± 0.2 | 44.1 ± 0.1 | 65.2 |
| MIRO (Cha et al., 2022) | I | 79.0 ± 0.0 | 85.4 ± 0.4 | 70.5 ± 0.4 | 50.4 ± 1.1 | 44.3 ± 0.2 | **65.9** |
| Ours (w/ Texts from dictionary) | I+T | 78.3 ± 0.4 | 85.7 ± 0.1 | 70.1 ± 0.1 | 49.5 ± 0.9 | 43.7 ± 0.0 | 65.5 |
| Ours (w/ Texts from InstructBLIP) | I+T | 78.6 ± 0.3 | 87.0 ± 0.4 | 70.4 ± 0.2 | 49.2 ± 0.5 | 44.2 ± 0.0 | **65.9** |

Figure 5 further shows the nearest examples in the unseen target domain to the given image (e.g., "dog"). In contrast to MIRO and GVRT, which often provide examples of different classes (e.g., "person"), ours consistently provide examples of the same class. This is consistent with our t-SNE analysis. We provide more examples in Appendix.

**Ablation Studies.** In Table 4, we conduct an ablation study to demonstrate the effect of main modules: (i) a global alignment, (ii) a local alignment, (iii) a visual graph, and (iv) a textual graph. Our study demonstrates that (1) a global alignment, which aligns graph-level features together, effectively improves accuracies, especially in photo, cartoon, and art domains. (2) Adding local alignment, which aligns graphs via the clustering-based matching algorithm, improves all domains while using both alignments outperforms the alternatives. (3) Either using a visual or textual graph alone improves model generalization, but the gain is marginal with the visual graph alone. (4) The gain is maximized by using both graphs, which indicates that a graph structure effectively transfers text knowledge to train a generalizable visual encoder.

**Analysis on DomainBed Benchmarks.** We also evaluate our model with a large-scale DomainBed (Gulrajani & Lopez-Paz, 2020) datasets. We use the following five multi-domain datasets, including VLCS (Fang et al., 2013), PACS (Li et al., 2017), OfficeHome (Venkateswara et al., 2017), TerraIncognita (Beery et al., 2018), and DomainNet (Peng et al., 2019), comparing ours with 19 domain generalization algorithms. Due to space constraints, we only report top-5 results (see Appendix for full table). Following GVRT (Min et al., 2022), we also report averaged results across three independent runs with randomly chosen hyperparameters. We observe in Table 5 that our proposed method shows matched or better state-of-the-art performance, where it ranks 1st (tied) in average performance.

## 5 CONCLUSION

This paper proposes a novel domain generalization method that encodes domain-invariant visual representations. To achieve this, we use a textual description to utilize verbalized (domain-invariant) knowledge from humans' typical reasoning. To align these, we use a clustering-based graph-matching algorithm based on visual and textual graphs built upon images and texts, respectively. We evaluate our model with state-of-the-art domain generalization approaches on CUB-DG and DomainBed datasets, achieving matched or higher scores than baselines.

## REPRODUCIBILITY STATEMENT

We provide detailed descriptions of our experimental settings and implementation details of our method in Section 4 and Appendix B. Additionally, we will release our code and data (i.e., our generated textual descriptions for DomainBed datasets) upon publication.

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

APPENDIX

In this appendix, we offer additional details that couldn't be accommodated in the main manuscript due to space limitations. Specifically, we provide a comprehensive exploration of the graph-based visual representation (Section A), implementation specifics (Section B), examples of descriptions generated using InstructBLIP (Section C), detailed t-SNE visualizations on CUB-DG (Section D), analysis of the PACS dataset (Section E), the full performance table for CUB-DG (Section F), and detailed DomainBed experiment results (Section G).

## A    GRAPH-BASED VISUAL REPRESENTATION

**Locally-aware Visual Graph Construction.**  In this section, we describe more detailed process of constructing the locally-aware visual graph. First, our backbone (ResNet50 (He et al., 2016b)) produces features of size $m' \times m' \times d$. The features are then transformed through average pooling to obtain a size of $m \times m \times d$, where $m \times m$ corresponds to $M$. This pooling operation is equivalent to dividing the image $\mathcal{I}$ into $M$ grids (refer to Figure 6 (a)). Each grid corresponds to a node in the visual graph, and possesses a $d$-dimensional feature. In our experiments, we set $m$ and $M$ to 14 and 196, respectively. Next, we compute the $L_2$ distance between each node and all other nodes in the graph, and sort them in ascending order. Subsequently, we select the $K_v$ nearest nodes to each node. Figure 6 (b) shows the process of ranking nodes based on their $L_2$ distance from each node, with only the top one nodes selected. (In Figure 6, We set $K_v$ to one for a brief explanation.) Finally, we can build the locally-aware visual graph, which has $M$ nodes with $K_v$ neighboring nodes.

**Graph-based Visual Representation.**  As described in the paper, we introduced an additional classifier that takes $\mathbf{g}_v$ as an input to effectively capture the class-discriminative features. This classifier is a linear layer trained by the standard cross-entropy loss. Analysis of the results presented in Table 6 shows the inferior performance when the aforementioned classifier is not trained, demonstrating that the classifier is crucial to performance.

Table 6: Out-of-distribution test accuracies (in %) on the CUB-DG dataset. We compare our model with and without graph-based visual representation classification.

| Classification for $\mathbf{g}_v$ | Target Domain | | | | Avg. |
|---|---|---|---|---|---|
| | Photo | Cartoon | Art | Paint | |
| - | 74.7 | 62.3 | 52.3 | 35.7 | 56.2 |
| ✓ | **75.4** | **65.5** | **54.0** | **41.5** | **59.1** |

## B    IMPLEMENTATION DETAILS

For the CUB-DG dataset, we configure the batch size to be 32 for each source domain. The value of $M$, which represents the number of local visual representations, varies depending on the input image dimensions. Specifically, we set $K_v$ and $K_t$ to 8 and 3, respectively, while $N_v$ and $N_t$ are both set to 5 and 3, respectively. In the context of the Domainbed benchmark, all hyperparameters, except for $\lambda$, are determined by the seed provided in Domainbed. We report the averaged results from three independent runs.

## C    DESCRIPTIONS GENERATED BY INSTRUCTBLIP

As mentioned earlier, domainbed benchmark datasets do not provide text inputs. To address this limitation, we employed InstructBLIP (Dai et al., 2023) for generating descriptions, specifying a maximum token limit of 30. Figure 7 shows examples of generated descriptions by InstructBLIP.

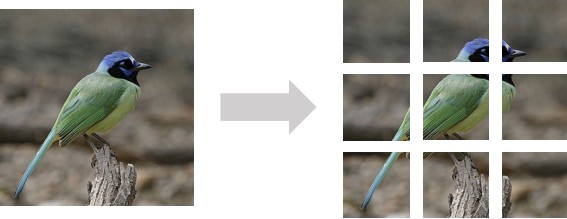

(a) Split the image into $M$ grids.

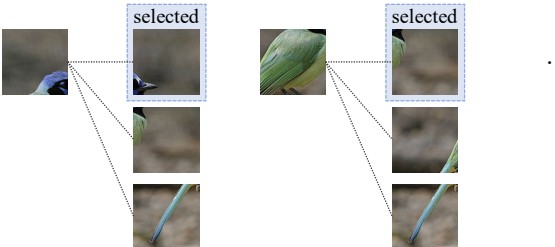

(b) Calculate the $L_2$ distance between each grid and select $K_v$ nearest nodes. In this figure, $K_v$ is set to one.

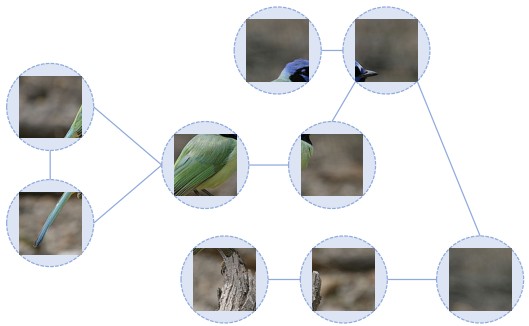

(c) Construct the graph based on steps (a) and (b)

Figure 6: 3 Steps to construct Locally-aware Visual Graph

# D    DETAILED T-SNE VISUALIZATIONS ON CUB-DG DATASET

In Figure 8, we provide a detailed t-SNE (Van der Maaten & Hinton, 2008) visualization of GVRT and ours with matched image samples. Note that we mark diffrent shapes to represent target domains and different colors to represent classes. In Figure 8 (a), images that belong to the same domain (ie. paint style) but diffrent classes are gathered together in the GVRT feature space. Examining the corresponding images, they have their own class discriminative characteristics like the color of beak and pattern of feather, except that they share a common domain style. In other words, the features of images can be located far away if the class discriminative characteristic is captured. Therefore, it can be inferred that the GVRT model relies more on the domain-specific features rather than domain-invaraint features for the images, limiting the ability of generalization. Figure 8 (b) shows the distribution of images that belong to the same class but different domains. In our model, the features of same classes are located close each other unlike GVRT where the features of paint domain are located far away. In fact, our inter-domain distance is lower than GVRT. Thus, we can infer that ours captures more domain-invariant features than GVRT for the images.

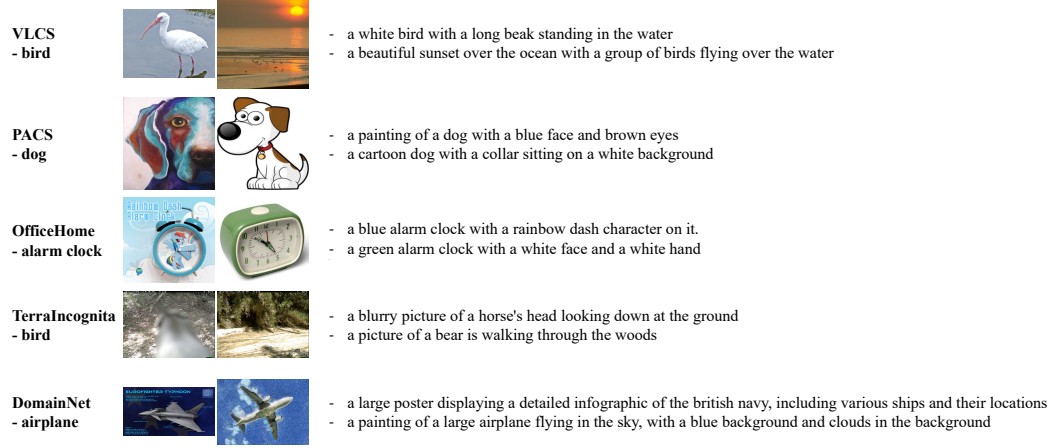

Figure 7: Example of a description generated by InstructBLIP for a randomly selected class and corresponding image for each dataset.

# E   ANALYSIS ON PACS DATASET

Figure 9 shows the top-2 nearest neighbors in the unseen target domain to the given images. In contrast to MIRO and GVRT, which often provide examples of different classes. In Figure 10, we present t-SNE visualizations for (a) MIRO, (b) GVRT, and (c) our model to illustrate their embedding spaces. In these visualizations, we generated these visualizations using 30 data samples for each class within each domain, employing distinct marker styles to represent target domains and various colors to distinguish between classes. Notably, both GVRT and our model outperform MIRO. Furthermore, when comparing our model to GVRT, we observe distinct improvements. Specifically, in the case of purple points representing the 'house' class and pink points corresponding to the 'person' class in the GVRT visualization, they are noticeably scattered and distant from each other. In contrast, our model exhibits significantly improved clustering, leading to a more compact and coherent distribution of data points.

Table 7: Full table for the out-of-distribution classification accuracies (in %) on CUB-DG dataset. *Abbr.* I: Image, T: Text.

| Model | Modality | Target Domain | | | | Avg. |
|---|---|---|---|---|---|---|
| | | Photo | Cartoon | Art | Paint | |
| IRM (Arjovsky et al., 2019) | I | 60.6 | 51.6 | 36.5 | 30.3 | 44.8 |
| GroupDRO (Sagawa et al., 2019) | I | 60.9 | 54.8 | 36.5 | 27.0 | 44.8 |
| ARM (Zhang et al., 2020) | I | 62.3 | 51.2 | 38.2 | 28.4 | 45.0 |
| ERM (Vapnik, 1999) | I | 62.5 | 53.2 | 37.4 | 29.0 | 45.5 |
| VREx (Krueger et al., 2020) | I | 63.9 | 54.9 | 38.6 | 30.1 | 46.9 |
| CDANN (Li et al., 2018c) | I | 65.3 | 55.2 | 43.2 | 30.5 | 48.6 |
| DANN (Ganin et al., 2016) | I | 67.5 | 57.0 | 42.8 | 30.6 | 49.5 |
| Mixup (Yan et al., 2020) | I | 67.1 | 55.9 | 51.1 | 27.2 | 50.3 |
| MixStyle (Zhou et al., 2020) | I | 59.0 | 56.7 | 50.3 | 35.8 | 50.4 |
| SagNet (Nam et al., 2021) | I | 67.4 | 60.7 | 44.0 | 34.2 | 51.6 |
| MIRO (Cha et al., 2022) | I | 68.2 | 59.1 | 46.5 | 38.2 | 53.0 |
| SD (Pezeshki et al., 2020) | I | 71.3 | 62.2 | 50.8 | 34.8 | 54.7 |
| CORAL (Sun & Saenko, 2016) | I | 72.2 | 63.5 | 50.3 | 35.8 | 55.4 |
| GVRT (Min et al., 2022) | I+T | 74.6 | 64.2 | 52.2 | 37.0 | 57.0 |
| Ours | I+T | **75.4** (0.8%↑) | **65.5** (1.4%↑) | **54.0** (1.8%↑) | **41.4** (4.4%↑) | **59.1** (2.1%↑) |

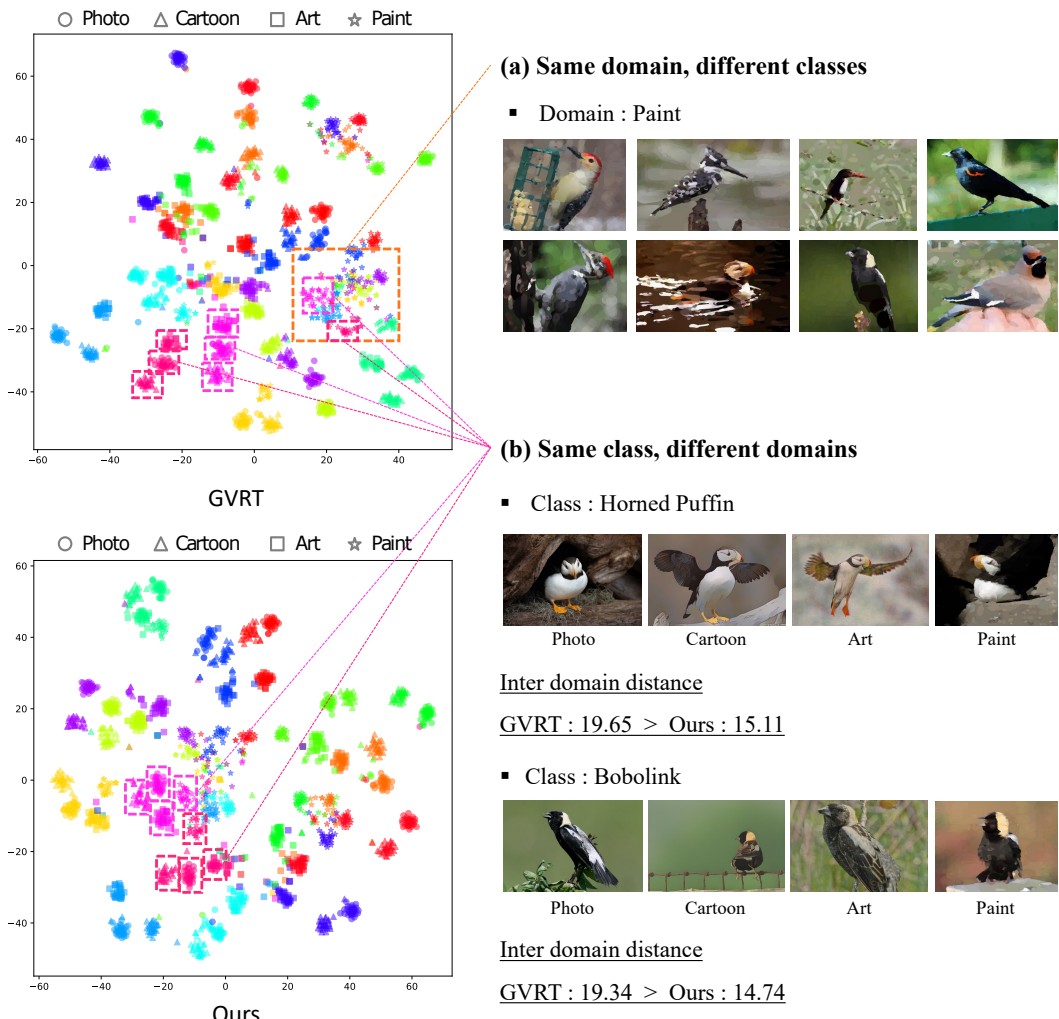

Figure 8: Visualization by t-SNE for GVRT and Ours with matched image samples.

## F    PEFORMANCE COMPARISON ON CUB-DG

In Table 7, we provide our experiment on the CUB-DG dataset and compare it with the following 14 existing domain generalization algorithms, including GVRT (Min et al., 2022), CORAL (Sun & Saenko, 2016), SD (Pezeshki et al., 2020), SagNet (Nam et al., 2021), MixStyle (Zhou et al., 2020), Mixup (Yan et al., 2020), DANN (Ganin et al., 2016), CDANN (Li et al., 2018c), VREx (Krueger et al., 2020), ERM (Vapnik, 1999), ARM (Zhang et al., 2020), GroupDRO (Sagawa et al., 2019), IRM (Arjovsky et al., 2019) and MIRO (Cha et al., 2022). The reported numbers for MIRO is the result of tuning the hyperparameters to suit the CUB-DG dataset, and the rest are all taken from GVRT.

## G    PERFORMANCE COMPARISON ON DOMAINBED

Table 8 is the full version of Table 5. In Table 9–13, we report per-domain results on each of the four multi-domain datasets from the large-scale DomainBed (Gulrajani & Lopez-Paz, 2020) experiments. We provide the averaged results from three independent trials. In each of the three trials, all choices, such as the dataset split, hyperparameter search, and weight initialization are selected randomly. For

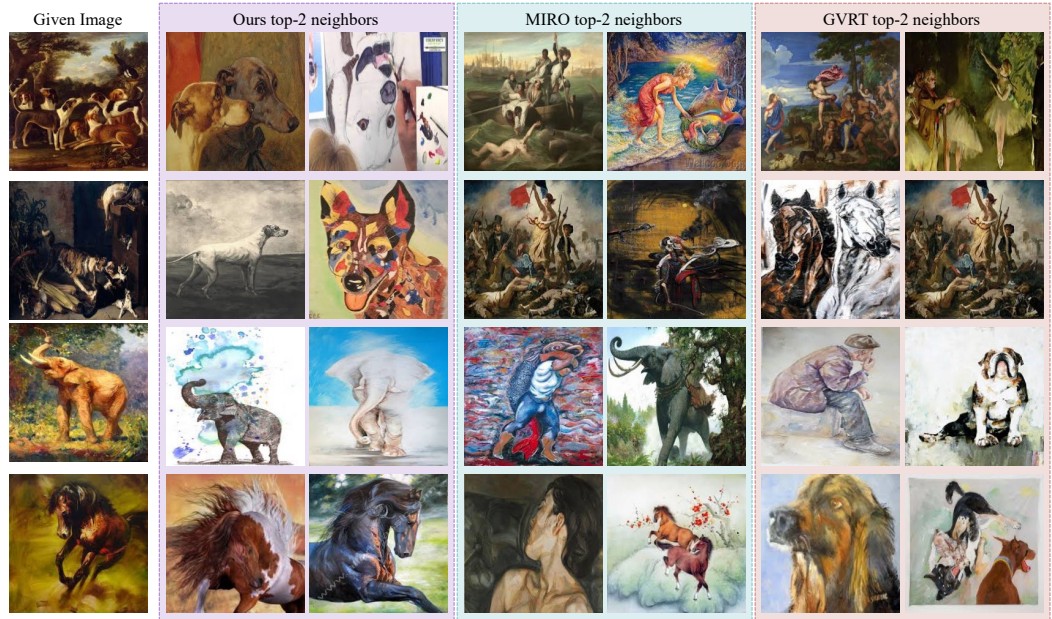

Figure 9: Exemplars of the top-2 nearest images to a specified image within the Art Painting domain of PACS dataset.

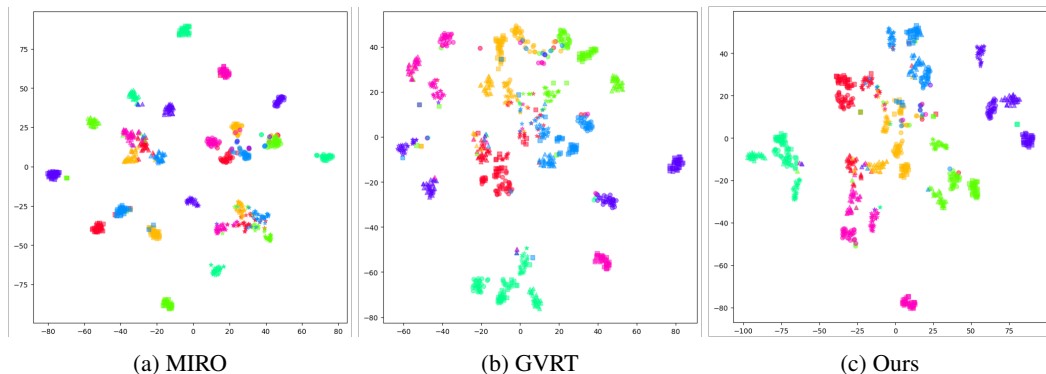

Figure 10: Visualizations by t-SNE for (a) MIRO, (b) GVRT and (c) Ours.

model selection, we used the validation set from the source domains. The reported numbers for SelfReg (Kim et al., 2021), and mDSI (Bui et al., 2021) were obtained from their respective papers, and the numbers for the remaining results were reported in the Domainbed (Gulrajani & Lopez-Paz, 2020). Note that GVRT (Min et al., 2022) and ours use multi-modal inputs (images and texts), while others only use images.

Table 8: Full table for the test accuracies (in %) on the Domainbed datasets in the multi-source DG task setting. *Abbr.* I: Image, T: Text.

| Algorithm | Modality | Dataset | | | | | Avg. |
|---|---|---|---|---|---|---|---|
| | | VLCS | PACS | OfficeHome | TerraIncognita | DomainNet | |
| Ours (w/ InstructBLIP) | I+T | 78.6 ± 0.3 | 87.0 ± 0.4 | 70.4 ± 0.2 | 49.2 ± 0.5 | 44.2 ± 0.0 | 65.9 |
| Ours (w/ dictionary) | I+T | 78.3 ± 0.4 | 85.7 ± 0.1 | 70.1 ± 0.1 | 49.5 ± 0.9 | 43.7 ± 0.0 | 65.5 |
| GVRT (PTE) (Min et al., 2022) | I+T | 79.0 ± 0.2 | 85.1 ± 0.3 | 70.1 ± 0.1 | 48.0 ± 0.2 | 44.1 ± 0.1 | 65.2 |
| MIRO (Cha et al., 2022) | I | 79.0 ± 0.0 | 85.4 ± 0.4 | 70.5 ± 0.4 | 50.4 ± 1.1 | 44.3 ± 0.2 | 65.9 |
| mDSDI (Bui et al., 2021) | I | 79.0 ± 0.3 | 86.2 ± 0.2 | 69.2 ± 0.4 | 48.1 ± 1.4 | 42.8 ± 0.1 | 65.1 |
| CORAL (Sun & Saenko, 2016) | I | 78.8 ± 0.6 | 86.2 ± 0.3 | 68.7 ± 0.3 | 47.6 ± 1.0 | 41.5 ± 0.1 | 64.6 |
| SagNet (Nam et al., 2021) | I | 77.8 ± 0.5 | 86.3 ± 0.2 | 68.1 ± 0.1 | 48.6 ± 1.0 | 40.3 ± 0.1 | 64.2 |
| SelfReg (Kim et al., 2021) | I | 77.8 ± 0.9 | 85.6 ± 0.4 | 67.9 ± 0.7 | 47.0 ± 0.3 | 42.8 ± 0.0 | 64.2 |
| Fish (Shi et al., 2021) | I | 77.8 ± 0.3 | 85.5 ± 0.3 | 68.6 ± 0.4 | 45.1 ± 1.3 | 42.7 ± 0.2 | 63.9 |
| MLDG (Li et al., 2018a) | I | 77.2 ± 0.4 | 84.9 ± 1.0 | 66.8 ± 0.6 | 47.7 ± 0.9 | 41.2 ± 0.1 | 63.6 |
| Mixup (Yan et al., 2020) | I | 77.4 ± 0.6 | 84.6 ± 0.6 | 68.1 ± 0.3 | 47.9 ± 0.8 | 39.2 ± 0.1 | 63.4 |
| ERM (Vapnik, 1999) | I | 77.5 ± 0.4 | 85.5 ± 0.2 | 66.5 ± 0.3 | 46.1 ± 1.8 | 40.9 ± 0.1 | 63.3 |
| MTL (Blanchard et al., 2021) | I | 77.2 ± 0.4 | 84.6 ± 0.5 | 66.4 ± 0.5 | 45.6 ± 1.2 | 40.6 ± 0.1 | 62.9 |
| RSC (Huang et al., 2020) | I | 77.1 ± 0.5 | 85.2 ± 0.9 | 65.5 ± 0.9 | 46.6 ± 1.0 | 38.9 ± 0.5 | 62.7 |
| DANN (Ganin et al., 2016) | I | 78.6 ± 0.4 | 83.6 ± 0.4 | 65.9 ± 0.6 | 46.7 ± 0.5 | 38.3 ± 0.1 | 62.6 |
| CDANN (Li et al., 2018c) | I | 77.5 ± 0.1 | 82.6 ± 0.9 | 65.8 ± 1.3 | 45.8 ± 1.6 | 38.3 ± 0.3 | 62.0 |
| VREx (Krueger et al., 2020) | I | 78.3 ± 0.2 | 84.9 ± 0.6 | 66.4 ± 0.6 | 46.4 ± 0.6 | 33.6 ± 2.9 | 61.9 |
| ARM (Zhang et al., 2020) | I | 77.6 ± 0.3 | 85.1 ± 0.4 | 64.8 ± 0.3 | 45.5 ± 0.3 | 35.5 ± 0.2 | 61.7 |
| IRM (Arjovsky et al., 2019) | I | 78.5 ± 0.5 | 83.5 ± 0.8 | 64.3 ± 2.2 | 47.6 ± 0.8 | 33.9 ± 2.8 | 61.6 |
| GroupDRO (Sagawa et al., 2019) | I | 76.7 ± 0.6 | 84.4 ± 0.8 | 66.0 ± 0.7 | 43.2 ± 1.1 | 33.3 ± 0.2 | 60.7 |
| MMD (Li et al., 2018b) | I | 77.5 ± 0.9 | 84.6 ± 0.5 | 66.3 ± 0.1 | 42.2 ± 1.6 | 23.4 ± 9.5 | 58.8 |

Table 9: Per-domain out-of-distribution test accuracies on the VLCS (Fang et al., 2013) dataset. *Abbr.* I: Image, T: Text

| Algorithm | Modality | Caltech | LabelMe | SUN09 | VOC2007 | Avg. |
|---|---|---|---|---|---|---|
| Ours (w/ InstructBLIP) | I+T | 98.3 ± 0.1 | 64.5 ± 0.3 | 73.7 ± 0.8 | 77.8 ± 1.1 | 78.6 |
| Ours (w/ dictionary) | I+T | 98.3 ± 0.3 | 64.6 ± 0.7 | 73.6 ± 2.2 | 76.6 ± 0.8 | 78.3 |
| GVRT (PTE) (Min et al., 2022) | I+T | 98.8 ± 0.1 | 64.0 ± 0.3 | 75.2 ± 0.5 | 77.9 ± 1.0 | 79.0 |
| MIRO (Cha et al., 2022) | I | - | - | - | - | 79.0 |
| mDSDI (Bui et al., 2021) | I | 97.6 ± 0.1 | 66.5 ± 0.4 | 74.0 ± 0.6 | 77.8 ± 0.7 | 79.0 |
| CORAL (Sun & Saenko, 2016) | I | 98.3 ± 0.1 | 66.1 ± 1.2 | 73.4 ± 0.3 | 77.5 ± 1.2 | 78.8 |
| DANN (Ganin et al., 2016) | I | 99.0 ± 0.3 | 65.1 ± 1.4 | 73.1 ± 0.3 | 77.2 ± 0.6 | 78.6 |
| IRM (Arjovsky et al., 2019) | I | 98.6 ± 0.1 | 64.9 ± 0.9 | 73.4 ± 0.6 | 77.3 ± 0.9 | 78.5 |
| VREx (Krueger et al., 2020) | I | 98.4 ± 0.3 | 64.4 ± 1.4 | 74.1 ± 0.4 | 76.2 ± 1.3 | 78.3 |
| SelfReg (Kim et al., 2021) | I | 96.7 ± 0.4 | 65.2 ± 1.2 | 73.1 ± 1.3 | 76.2 ± 0.7 | 77.8 |
| SagNet (Nam et al., 2021) | I | 97.9 ± 0.4 | 64.5 ± 0.5 | 71.4 ± 1.3 | 77.5 ± 0.5 | 77.8 |
| Fish (Shi et al., 2021) | I | - | - | - | - | 77.8 |
| ARM (Zhang et al., 2020) | I | 98.7 ± 0.2 | 63.6 ± 0.7 | 71.3 ± 1.2 | 76.7 ± 0.6 | 77.6 |
| MMD (Li et al., 2018b) | I | 97.7 ± 0.1 | 64.0 ± 1.1 | 72.8 ± 0.2 | 75.3 ± 3.3 | 77.5 |
| CDANN (Li et al., 2018c) | I | 97.1 ± 0.3 | 65.1 ± 1.2 | 70.7 ± 0.8 | 77.1 ± 1.5 | 77.5 |
| ERM (Vapnik, 1999) | I | 97.7 ± 0.4 | 64.3 ± 0.9 | 73.4 ± 0.5 | 74.6 ± 1.3 | 77.5 |
| Mixup (Yan et al., 2020) | I | 98.3 ± 0.6 | 64.8 ± 1.0 | 72.1 ± 0.5 | 74.3 ± 0.8 | 77.4 |
| MTL (Blanchard et al., 2021) | I | 97.8 ± 0.4 | 64.3 ± 0.3 | 71.5 ± 0.7 | 75.3 ± 1.7 | 77.2 |
| MLDG (Li et al., 2018a) | I | 97.4 ± 0.2 | 65.2 ± 0.7 | 71.0 ± 1.4 | 75.3 ± 1.0 | 77.2 |
| RSC (Huang et al., 2020) | I | 97.9 ± 0.1 | 62.5 ± 0.7 | 72.3 ± 1.2 | 75.6 ± 0.8 | 77.1 |
| GroupDRO (Sagawa et al., 2019) | I | 97.3 ± 0.3 | 63.4 ± 0.9 | 69.5 ± 0.8 | 76.7 ± 0.7 | 76.7 |

Table 10: Per-domain out-of-distribution test accuracies on the PACS (Li et al., 2017) dataset. *Abbr.* I: Image, T: Text

| Algorithm | Modality | Art Painting | Cartoon | Photo | Sketch | Avg. |
|---|---|---|---|---|---|---|
| Ours (w/ InstructBLIP) | I+T | 87.9 ± 0.7 | 81.4 ± 0.1 | 98 ± 0.1 | 80.5 ± 1.1 | 87.0 |
| Ours (w/ dictionary) | I+T | 87.1 ± 0.5 | 79.8 ± 0.4 | 97.7 ± 0.1 | 78.3 ± 0.7 | 85.7 |
| GVRT (PTE)  (Min et al., 2022) | I+T | 87.9 ± 0.3 | 78.4 ± 1.0 | 98.2 ± 0.1 | 75.7 ± 0.4 | 85.1 |
| SagNet (Nam et al., 2021) | I | 87.4 ± 1.0 | 80.7 ± 0.6 | 97.1 ± 0.1 | 80.0 ± 0.4 | 86.3 |
| mDSDI (Bui et al., 2021) | I | 87.7 ± 0.4 | 80.4 ± 0.7 | 98.1 ± 0.3 | 78.4 ± 1.2 | 86.2 |
| CORAL (Sun & Saenko, 2016) | I | 88.3 ± 0.2 | 80.0 ± 0.5 | 97.5 ± 0.3 | 78.8 ± 1.3 | 86.2 |
| SelfReg (Kim et al., 2021) | I | 87.9 ± 1.0 | 79.4 ± 1.4 | 96.8 ± 0.7 | 78.3 ± 1.2 | 85.6 |
| ERM (Vapnik, 1999) | I | 84.7 ± 0.4 | 80.8 ± 0.6 | 97.2 ± 0.3 | 79.3 ± 1.0 | 85.5 |
| Fish (Shi et al., 2021) | I | - | - | - | - | 85.5 |
| MIRO (Cha et al., 2022) | I | - | - | - | - | 85.4 |
| RSC (Huang et al., 2020) | I | 85.4 ± 0.8 | 79.7 ± 1.8 | 97.6 ± 0.3 | 78.2 ± 1.2 | 85.2 |
| ARM (Zhang et al., 2020) | I | 86.8 ± 0.6 | 76.8 ± 0.5 | 97.4 ± 0.3 | 79.3 ± 1.2 | 85.1 |
| VREx (Krueger et al., 2020) | I | 86.0 ± 1.6 | 79.1 ± 0.6 | 96.9 ± 0.5 | 77.7 ± 1.7 | 84.9 |
| MLDG (Li et al., 2018a) | I | 85.5 ± 1.4 | 80.1 ± 1.7 | 97.4 ± 0.3 | 76.6 ± 1.1 | 84.9 |
| MMD (Li et al., 2018b) | I | 86.1 ± 1.4 | 79.4 ± 0.9 | 96.6 ± 0.2 | 76.5 ± 0.5 | 84.6 |
| MTL (Blanchard et al., 2021) | I | 87.5 ± 0.8 | 77.1 ± 0.5 | 96.4 ± 0.8 | 77.3 ± 1.8 | 84.6 |
| Mixup (Yan et al., 2020) | I | 86.1 ± 0.5 | 78.9 ± 0.8 | 97.6 ± 0.1 | 75.8 ± 1.8 | 84.6 |
| GroupDRO (Sagawa et al., 2019) | I | 83.5 ± 0.9 | 79.1 ± 0.6 | 96.7 ± 0.3 | 78.3 ± 2.0 | 84.4 |
| DANN (Ganin et al., 2016) | I | 86.4 ± 0.8 | 77.4 ± 0.8 | 97.3 ± 0.4 | 73.5 ± 2.3 | 83.6 |
| IRM (Arjovsky et al., 2019) | I | 84.8 ± 1.3 | 76.4 ± 1.1 | 96.7 ± 0.6 | 76.1 ± 1.0 | 83.5 |
| CDANN (Li et al., 2018c) | I | 84.6 ± 1.8 | 75.5 ± 0.9 | 96.8 ± 0.3 | 73.5 ± 0.6 | 82.6 |

Table 11: Per-domain out-of-distribution test accuracies on the OfficeHome (Venkateswara et al., 2017) dataset. *Abbr.* I: Image, T: Text

| Algorithm | Modality | Art | Clipart | Product | Real-world | Avg. |
|---|---|---|---|---|---|---|
| Ours (w/ InstructBLIP) | I+T | 66.5 ± 0.4 | 56.4 ± 0.4 | 78.5 ± 0.5 | 80.1 ± 0.1 | 70.4 |
| Ours (w/ dictionary) | I+T | 66.7 ± 1.0 | 55.4 ± 0.4 | 78.2 ± 0.4 | 80.0 ± 0.3 | 70.1 |
| GVRT (PTE)  (Min et al., 2022) | I+T | 66.3 ± 0.1 | 55.8 ± 0.4 | 78.2 ± 0.4 | 80.4 ± 0.2 | 70.1 |
| MIRO (Cha et al., 2022) | I | - | - | - | - | 70.5 |
| mDSDI (Bui et al., 2021) | I | 68.1 ± 0.3 | 52.1 ± 0.4 | 76.0 ± 0.2 | 80.4 ± 0.2 | 69.2 |
| CORAL (Sun & Saenko, 2016) | I | 65.3 ± 0.4 | 54.4 ± 0.5 | 76.5 ± 0.1 | 78.4 ± 0.5 | 68.7 |
| Fish (Shi et al., 2021) | I | - | - | - | - | 68.6 |
| Mixup (Yan et al., 2020) | I | 62.4 ± 0.8 | 54.8 ± 0.6 | 76.9 ± 0.3 | 78.3 ± 0.2 | 68.1 |
| SagNet (Nam et al., 2021) | I | 63.4 ± 0.2 | 54.8 ± 0.4 | 75.8 ± 0.4 | 78.3 ± 0.3 | 68.1 |
| SelfReg (Kim et al., 2021) | I | 63.6 ± 1.4 | 53.1 ± 1.0 | 76.9 ± 0.4 | 78.1 ± 0.4 | 67.9 |
| MLDG (Li et al., 2018a) | I | 61.5 ± 0.9 | 53.2 ± 0.6 | 75.0 ± 1.2 | 77.5 ± 0.4 | 66.8 |
| ERM (Vapnik, 1999) | I | 61.3 ± 0.7 | 52.4 ± 0.3 | 75.8 ± 0.1 | 76.6 ± 0.3 | 66.5 |
| MTL (Blanchard et al., 2021) | I | 61.5 ± 0.7 | 52.4 ± 0.6 | 74.9 ± 0.4 | 76.8 ± 0.4 | 66.4 |
| VREx (Krueger et al., 2020) | I | 60.7 ± 0.9 | 53.0 ± 0.9 | 75.3 ± 0.1 | 76.6 ± 0.5 | 66.4 |
| MMD (Li et al., 2018b) | I | 60.4 ± 0.2 | 53.3 ± 0.3 | 74.3 ± 0.1 | 77.4 ± 0.6 | 66.3 |
| GroupDRO (Sagawa et al., 2019) | I | 60.4 ± 0.7 | 52.7 ± 1.0 | 75.0 ± 0.7 | 76.0 ± 0.7 | 66.0 |
| DANN (Ganin et al., 2016) | I | 59.9 ± 1.3 | 53.0 ± 0.3 | 73.6 ± 0.7 | 76.9 ± 0.5 | 65.9 |
| CDANN (Li et al., 2018c) | I | 61.5 ± 1.4 | 50.4 ± 2.4 | 74.4 ± 0.9 | 76.6 ± 0.8 | 65.8 |
| RSC (Huang et al., 2020) | I | 60.7 ± 1.4 | 51.4 ± 0.3 | 74.8 ± 1.1 | 75.1 ± 1.3 | 65.5 |
| ARM (Zhang et al., 2020) | I | 58.9 ± 0.8 | 51.0 ± 0.5 | 74.1 ± 0.1 | 75.2 ± 0.3 | 64.8 |
| IRM (Arjovsky et al., 2019) | I | 58.9 ± 2.3 | 52.2 ± 1.6 | 72.1 ± 2.9 | 74.0 ± 2.5 | 64.3 |

Table 12: Per-domain out-of-distribution test accuracies on the TerraIncognita (Beery et al., 2018) dataset. *Abbr.* I: Image, T: Text

| Algorithm | Modality | L100 | L38 | L43 | L46 | Avg. |
|---|---|---|---|---|---|---|
| Ours (w/ InstructBLIP) | I+T | $54.5 \pm 0.6$ | $46.7 \pm 0.8$ | $57.1 \pm 1.2$ | $39 \pm 0.8$ | 49.2 |
| Ours (w/ dictionary) | I+T | $56.9 \pm 3.0$ | $45.5 \pm 0.7$ | $57.7 \pm 1.4$ | $37.8 \pm 0.8$ | 49.5 |
| GVRT (PTE) (Min et al., 2022) | I+T | $53.9 \pm 1.3$ | $41.8 \pm 1.2$ | $58.2 \pm 0.9$ | $38.0 \pm 0.6$ | 48.0 |
| MIRO (Cha et al., 2022) | I | - | - | - | - | 50.4 |
| SagNet (Nam et al., 2021) | I | $53.0 \pm 2.9$ | $43.0 \pm 2.5$ | $57.9 \pm 0.6$ | $40.4 \pm 1.3$ | 48.6 |
| mDSDI (Bui et al., 2021) | I | $53.2 \pm 3.0$ | $43.3 \pm 1.0$ | $56.7 \pm 0.5$ | $39.2 \pm 1.3$ | 48.1 |
| Mixup (Yan et al., 2020) | I | $59.6 \pm 2.0$ | $42.2 \pm 1.4$ | $55.9 \pm 0.8$ | $33.9 \pm 1.4$ | 47.9 |
| MLDG (Li et al., 2018a) | I | $54.2 \pm 3.0$ | $44.3 \pm 1.1$ | $55.6 \pm 0.3$ | $36.9 \pm 2.2$ | 47.7 |
| IRM (Arjovsky et al., 2019) | I | $54.6 \pm 1.3$ | $39.8 \pm 1.9$ | $56.2 \pm 1.8$ | $39.6 \pm 0.8$ | 47.6 |
| CORAL (Sun & Saenko, 2016) | I | $51.6 \pm 2.4$ | $42.2 \pm 1.0$ | $57.0 \pm 1.0$ | $39.8 \pm 2.9$ | 47.6 |
| SelfReg (Kim et al., 2021) | I | $48.8 \pm 0.9$ | $41.3 \pm 1.8$ | $57.3 \pm 0.7$ | $40.6 \pm 0.9$ | 47.0 |
| DANN (Ganin et al., 2016) | I | $51.1 \pm 3.5$ | $40.6 \pm 0.6$ | $57.4 \pm 0.5$ | $37.7 \pm 1.8$ | 46.7 |
| RSC (Huang et al., 2020) | I | $50.2 \pm 2.2$ | $39.2 \pm 1.4$ | $56.3 \pm 1.4$ | $40.8 \pm 0.6$ | 46.6 |
| VREx (Krueger et al., 2020) | I | $48.2 \pm 4.3$ | $41.7 \pm 1.3$ | $56.8 \pm 0.8$ | $38.7 \pm 3.1$ | 46.4 |
| ERM (Vapnik, 1999) | I | $49.8 \pm 4.4$ | $42.1 \pm 1.4$ | $56.9 \pm 1.8$ | $35.7 \pm 3.9$ | 46.1 |
| CDANN (Li et al., 2018c) | I | $47.0 \pm 1.9$ | $41.3 \pm 4.8$ | $54.9 \pm 1.7$ | $39.8 \pm 2.3$ | 45.8 |
| MTL (Blanchard et al., 2021) | I | $49.3 \pm 1.2$ | $39.6 \pm 6.3$ | $55.6 \pm 1.1$ | $37.8 \pm 0.8$ | 45.6 |
| ARM (Zhang et al., 2020) | I | $49.3 \pm 0.7$ | $38.3 \pm 2.4$ | $55.8 \pm 0.8$ | $38.7 \pm 1.3$ | 45.5 |
| Fish (Shi et al., 2021) | I | - | - | - | - | 45.1 |
| GroupDRO (Sagawa et al., 2019) | I | $41.2 \pm 0.7$ | $38.6 \pm 2.1$ | $56.7 \pm 0.9$ | $36.4 \pm 2.1$ | 43.2 |
| MMD (Li et al., 2018b) | I | $41.9 \pm 3.0$ | $34.8 \pm 1.0$ | $57.0 \pm 1.9$ | $35.2 \pm 1.8$ | 42.2 |

Table 13: Per-domain out-of-distribution test accuracies on the DomainNet (Peng et al., 2019) dataset. *Abbr.* I: Image, T: Text

| Algorithm | Modality | Clip | Info | Paint | Quick | Real | Sketch | Avg. |
|---|---|---|---|---|---|---|---|---|
| Ours (w/ InstructBLIP) | I+T | $61.6 \pm 0.2$ | $21.1 \pm 0.1$ | $51.3 \pm 0.1$ | $13.9 \pm 0.2$ | $64.8 \pm 0.1$ | $52.5 \pm 0.2$ | 44.2 |
| Ours (w/ dictionary) | I+T | $61.1 \pm 0.1$ | $20.4 \pm 0.2$ | $50.4 \pm 0.1$ | $13.5 \pm 0.1$ | $64.7 \pm 0.3$ | $51.9 \pm 0.1$ | 43.7 |
| GVRT (PTE) (Min et al., 2022) | I+T | $62.4 \pm 0.4$ | $21.0 \pm 0.0$ | $50.5 \pm 0.4$ | $13.8 \pm 0.3$ | $64.6 \pm 0.4$ | $52.4 \pm 0.2$ | 44.1 |
| mDSDI (Bui et al., 2021) | I | $62.1 \pm 0.3$ | $19.1 \pm 0.4$ | $49.4 \pm 0.4$ | $12.8 \pm 0.7$ | $62.9 \pm 0.3$ | $50.4 \pm 0.4$ | 42.8 |
| CORAL (Sun & Saenko, 2016) | I | $59.2 \pm 0.1$ | $19.7 \pm 0.2$ | $46.6 \pm 0.3$ | $13.4 \pm 0.4$ | $59.8 \pm 0.2$ | $50.1 \pm 0.6$ | 41.5 |
| SagNet (Nam et al., 2021) | I | $57.7 \pm 0.3$ | $19.0 \pm 0.2$ | $45.3 \pm 0.3$ | $12.7 \pm 0.5$ | $58.1 \pm 0.5$ | $48.8 \pm 0.2$ | 40.3 |
| SelfReg (Kim et al., 2021) | I | $60.7 \pm 0.1$ | $21.6 \pm 0.1$ | $49.4 \pm 0.2$ | $12.7 \pm 0.1$ | $60.7 \pm 0.1$ | $51.7 \pm 0.1$ | 42.8 |
| Mixup (Yan et al., 2020) | I | $55.7 \pm 0.3$ | $18.5 \pm 0.5$ | $44.3 \pm 0.5$ | $12.5 \pm 0.4$ | $55.8 \pm 0.3$ | $48.2 \pm 0.5$ | 39.2 |
| MLDG (Li et al., 2018a) | I | $59.1 \pm 0.2$ | $19.1 \pm 0.3$ | $45.8 \pm 0.7$ | $13.4 \pm 0.3$ | $59.6 \pm 0.2$ | $50.2 \pm 0.4$ | 41.2 |
| VREx (Krueger et al., 2020) | I | $47.3 \pm 3.5$ | $16.0 \pm 1.5$ | $35.8 \pm 4.6$ | $10.9 \pm 0.3$ | $49.6 \pm 4.9$ | $42.0 \pm 3.0$ | 33.6 |
| ERM (Vapnik, 1999) | I | $58.1 \pm 0.3$ | $18.8 \pm 0.3$ | $46.7 \pm 0.3$ | $12.2 \pm 0.4$ | $59.6 \pm 0.1$ | $49.8 \pm 0.4$ | 40.9 |
| DANN (Ganin et al., 2016) | I | $53.1 \pm 0.2$ | $18.3 \pm 0.1$ | $44.2 \pm 0.7$ | $11.8 \pm 0.1$ | $55.5 \pm 0.4$ | $46.8 \pm 0.6$ | 38.3 |
| RSC (Huang et al., 2020) | I | $55.0 \pm 1.2$ | $18.3 \pm 0.5$ | $44.4 \pm 0.6$ | $12.2 \pm 0.2$ | $55.7 \pm 0.7$ | $47.8 \pm 0.9$ | 38.9 |
| IRM (Arjovsky et al., 2019) | I | $48.5 \pm 2.8$ | $15.0 \pm 1.5$ | $38.3 \pm 4.3$ | $10.9 \pm 0.5$ | $48.2 \pm 5.2$ | $42.3 \pm 3.1$ | 33.9 |
| MTL (Blanchard et al., 2021) | I | $57.9 \pm 0.5$ | $18.5 \pm 0.4$ | $46.0 \pm 0.1$ | $12.5 \pm 0.1$ | $59.5 \pm 0.3$ | $49.2 \pm 0.1$ | 40.6 |
| ARM (Zhang et al., 2020) | I | $49.7 \pm 0.3$ | $16.3 \pm 0.5$ | $40.9 \pm 1.1$ | $9.4 \pm 0.1$ | $53.4 \pm 0.4$ | $43.5 \pm 0.4$ | 35.5 |
| CDANN (Li et al., 2018c) | I | $54.6 \pm 0.4$ | $17.3 \pm 0.1$ | $43.7 \pm 0.9$ | $12.1 \pm 0.7$ | $56.2 \pm 0.4$ | $45.9 \pm 0.5$ | 38.3 |
| MMD (Li et al., 2018b) | I | $32.1 \pm 13.3$ | $11.0 \pm 4.6$ | $26.8 \pm 11.3$ | $8.7 \pm 2.1$ | $32.7 \pm 13.8$ | $28.9 \pm 11.9$ | 23.4 |
| GroupDRO (Sagawa et al., 2019) | I | $47.2 \pm 0.5$ | $17.5 \pm 0.4$ | $33.8 \pm 0.5$ | $9.3 \pm 0.3$ | $51.6 \pm 0.4$ | $40.1 \pm 0.6$ | 33.3 |
| MIRO (Cha et al., 2022) | I | - | - | - | - | - | - | 44.3 |
| Fish (Shi et al., 2021) | I | - | - | - | - | - | - | 42.7 |

