# OpenReview forum: "Bridging the Domain Gap by Clustering-based Image-Text Graph Matching"
_ICLR.cc/2024/Conference — ICLR 2024 Conference Withdrawn Submission_

### Official Review · Reviewer_kUjy · 2023-10-25

**Soundness:** 2 fair
**Presentation:** 2 fair
**Contribution:** 2 fair
**Rating:** 3
**Confidence:** 5

**Summary:**

This paper proposes a multimodal graph representation with textual semantic cues to obtain domain-invariant pivot embeddings for the domain generalization problem. The authors utilize graph neural network method to separately represent image and text descriptions, and cluster and match graph node features to learn domain-invariant features. The experiments verify the effectiveness of the proposed method.

**Strengths:**

1)	This paper proposes a simple and effective image-text graph matching method to solve the domain generalization problem.

2)	Experimental results show that the proposed method achieves state-of-the-art performance on domain generalization tasks

**Weaknesses:**

1)	The novelty of the proposed method is limited. Compared with GVRT [1], the proposed method only changes the manner of image-text alignment (adding graph), and thus provides incremental novelty. Moreover, graph-based image-text matching has been broadly studied [2-4], and the proposed technique contributes very poorly.

[1] Min S, Park N, Kim S, et al. Grounding visual representations with texts for domain generalization. In ECCV, 2022
[2] Liu C, Mao Z, Zhang T, et al. Graph structured network for image-text matching. In CVPR, 2020
[3] Wang S, Wang R, Yao Z, et al. Cross-modal scene graph matching for relationship-aware image-text retrieval. In WACV, 2020
[4] Li Y, Zhang D, Mu Y. Visual-semantic matching by exploring high-order attention and distraction. In CVPR, 2020

2)	The construction mechanism of the two graphs (visual and textual graphs) is unclear, especially the connection relations between graph edges.

3)	Why graph matching contribute to learning domain-invariant features? The authors should provide detailed explanations of the theoretical basis of the proposed framework and provide more insights.

4)	There are many hyper-parameters (more than six), which need to be reported empirically via curve charts.

5)	The comparison of methods is not particularly sufficient. The latest comparative literature is from 2022, and there is a lack of comparison of related work as recently as 2023.

**Questions:**

Please see the weakness.

---

> ### Author Response · Authors · 2023-11-14
> **Reply to your valuable comments [1/3]**
>
> We would like to express our sincere appreciation for your time and expertise in reviewing our paper. We are grateful for the thoroughness of your review.
>
> > **W1. The novelty of the proposed method is limited. Compared with GVRT [1], the proposed method only changes the manner of image-text alignment (adding graph), and thus provides incremental novelty. Moreover, graph-based image-text matching has been broadly studied [2-4], and the proposed technique contributes very poorly.**
>
> **A1.** In contrast to GVRT's method of aligning a single image feature with a corresponding text feature using L2 distance as the loss, our model introduces an additional layer of complexity through a dual alignment strategy. Our method involves both global and local alignment processes. For global alignment, we align the entire image and text features. Simultaneously, we incorporate local alignment by partitioning the image into multiple grids and treating each grid as an individual feature. Similarly, the text is divided into several words, with each word treated as an independent feature. Visual graph and textual graph is then constructed to facilitate alignment at the local level.
>
>  This dual alignment strategy enables our model to capture local information that might be overlooked in a purely global alignment framework. By considering both the broader context and finer local details, we enhance the alignment between images and text, consequently improving performance in domain generalization. In fact, as shown in the attatched figure **[(image link)](https://drive.google.com/file/d/11l39-Bv4sPmT8UI1aufzyvnbd1wasYPK/view?usp=drive_link)**, you can see that our model captures more diverse features than GVRT.
>
>  As you rightly pointed out, while utilizing graph models to represent visual and textual information and performing graph matching is a common approach in image-text retrieval, to the best of my knowledge, this research is the first to apply these techniques specifically in the domain generalization task. Our method leverages graph models to represent both visual and textual information, achieving improved performance in Domain Generalization (DG) tasks. Specifically, by using image-text information from the source (seen) domain, our model demonstrates strong performance even in target (unseen) domains with entirely different data distributions. This is substantiated by our outstanding results on the CUB-DG dataset (Upper table in Table 1) and the Domainbed benchmark (Table 5).
>
>  And as you pointed out, it may initially appear that the improvement on DomainBed is not substantial. In CUB-DG dataset, where our model surpasses the ERM [1] by 13.6%. In the context of DomainBed, the performance disparity between our model (SOTA) and ERM is 2.6%. It's worth noting that in DomainBed, the absolute performance change may not be as pronounced. For example, Table 1 shows the top five methods in DomainBed's PACS dataset. It is meaningful that the performance difference between our model and the existing sota is 0.7%, compared to the 0.1% performance difference between the existing sota, SagNet [2], and the previous sota, mDSDI [3].  Despite this,our model attains the SOTA in both CUB-DG and DomainBed, showcasing its effectiveness across various settings. Moreover, our model consistently outperforms other methods, even in challenging few-shot scenarios. (Table 2)
>
> > **W3. Why graph matching contribute to learning domain-invariant features? The authors should provide detailed explanations of the theoretical basis of the proposed framework and provide more insights.**
>
> **A3.** Similar to Answer 1,  leveraging dual alignment strategy enables our model to capture local information that might be overlooked in a purely global alignment framework. By considering both the broader context and finer local details, we enhance the alignment between images and text, consequently improving performance in domain generalization. In fact, as shown in the attached figure in Answer 1 (GradCAM visualization), you can see that our model captures more diverse features than GVRT.
>
>
> [1] Vladimir N Vapnik. An overview of statistical learning theory. IEEE transactions on neural networks, 10(5):988–999, 1999
> [2] Hyeonseob Nam et al. Reducing domain gap by reducing style bias. In CVPR, 2021
> [3] Manh-Ha Bui, Toan Tran, Anh Tran, and Dinh Phung. Exploiting domain-specific features to enhance domain generalization. Advances in Neural Information Processing Systems, 34:21189–
> 21201, 2021.

---

> ### Author Response · Authors · 2023-11-14
> **Reply to your valuable comments [2/3]**
>
> > **W2. The construction mechanism of the two graphs (visual and textual graphs) is unclear, especially the connection relations between graph edges.**
>
> **A2.** We appreciate your feedback, and we apologize for any confusion in the explanation. Allow me to provide a more detailed clarification regarding the process of constructing the graphs:
> - Visual Graph
> 	- When an image of $\mathbb{R}^{H \times W \times 3}$ dimensions enters the input of ResNet50 (backbone), a vector of $\mathbb{R}^{m \times m \times d}$ dimensions is generated through four resnet blocks, and after that a vector of $\mathbb{R}^d$ dimensions is generated through the global average pooling layer. At this time, the vector of the d-dimensional becomes global visual representation, and a visual graph is built from a vector of $\mathbb{R}^{m \times m \times d}$ dimensions.
> 	- The $d$-dimensional vector serves as the global visual representation, and the $m \times m \times d$-dimensional vector is used to form the visual graph.
> 	- As for the vectors in the $m \times m \times d$-dimension, it can be seen that there are $M$ local visual representations which dimension is $d$. ($M = m \times m$)
> 	- For each local visual representation, we calculate the distance (L2) between itself and the remaining $M$-1 local visual representations. We then consider the $K_v$ closest neighbors as adjacent nodes, assuming edges between these neighbors and the central node. This results in a total of $m \times (m - 1)$ edges.
> 	- For a figure of this process, please refer to Figure 6 in the Appendix.
>
> - Textual Graph
> 	- Similar to Visual Graph.
> 	- In the Visual Graph, each feature of $m \times m \times d$-dimension becomes a local visual feature before passing through resnet50's global average pooling layer, while in the Textual Graph, each word embedding becomes a local textual feature.
> 	- Similar to the Visual Graph, we compute the L2 distance to all other local textual features except itself, take the nearest $K_t$ local textual features as neighbors, and connect the edges.
>
>  We hope this detailed explanation clarifies the construction mechanism of the graphs, especially the connection relations between graph edges. If you have any further questions or concerns, please feel free to inquire.
>
> > **W4. There are many hyper-parameters (more than six), which need to be reported empirically via curve charts.**
>
> **A4.** Indeed, as you pointed out, the paper provides many hyperparameters requiring adjustments in our model. It was my intention to convey to the readers that these parameters are subject to their control. However, among the hyperparameters outlined in the paper, we set fixed values for lambda (both lambda_global and lambda_local were set to 1, and lambda_p, lambda_h, lambda_d, lambda_aux were set to 0.1, 0.1, 1, 1, respectively). Thus, the practical hyperparameters involve determining the number of neighbor nodes (K_v, K_t) during the construction of visual and textual graphs, as well as deciding the number of clusters (N_v, N_t) in the graph node clustering process following graph creation.
>
>  In addition, although there are many hyperparameters, results on DomainBed (Table 5) show that the performance is averaged after randomly turning it into a random hyperparameter through three seeds, so it has robustness that does not fall below a certain level of performance.
>
> - **Number of neighbor nodes.** We conducted experiments on CUB-DG datasets according to $K_v$ and $K_t$. And we observed a tension between different combinations, and we used the best setting as default: ($K_v$, $K_t$) = (8, 3).
>
> | $K_v$ | $K_t$ | Photo | Cartoon | Art | Paint | Avg. |
> |---|---|---|---|---|---|---|
> | 8 | 3 | 75.3 | 65.5 | 54.0 | 41.4 | 59.1 |
> | 8 | 1 | 75.5 | 65.0 | 53.3 | 38.3 | 58.0 |
> | 8 | 5 |75.7 |64.1 | 54.2 | 40.9 | 58.7|
> | 4 | 3 | 75.3| 64.9 |53.5 |35.4 |57.3|
> | 24 | 3 |75.0| 66.0 |53.7 |37.9 |58.2|
>
> - **Number of cluster.** We also conducted experiments according to $N_v$ and $N_t$ on the CUB-DG dataset. (This experiment was conducted on a gpu different from previous experiments because of the resource issue, thus there is a slight difference in performance due to this.) Looking at the table below, the difference in performance according to $N_v$ and $N_t$ was not large. However, when ($N_v$, $N_t$)=(5,5), it showed the best performance, but since the length of the text is not that long, if $N_t$ is set to 5, a meaningless word cluster is formed, so we used ($N_v$, $N_t$)=(5, 3) as default.
>
> | $N_v$ | $N_t$ | Photo | Cartoon | Art | Paint | Avg. |
> |---|---|---|---|---|---|---|
> |5 |3|75.73|66.03|54.33|39.25 | 58.8|
> |5 | 4| 74.78|65.52|53.88|40.44|58.7|
> | 5|5|75.58|66.05|54.75|39.77|59.0|
> |5|8|75.03|65.95|55.07|40.3|59.1|
> | 6 | 6 | 75.49|65.26|53.33|40.13|58.6|

---

> ### Author Response · Authors · 2023-11-14
> **Reply to your valuable comments [3/3]**
>
> > **W5. The comparison of methods is not particularly sufficient. The latest comparative literature is from 2022, and there is a lack of comparison of related work as recently as 2023.**
>
> **A5.** As far as we are aware, MIRO is the most recent and SOTA paper in this setting. To ensure a comprehensive comparison, we have included experimental results that encompass MIRO. However, if there are other recent methods that you suggest we consider for comparison, please provide the references, and we will gladly incorporate them into our evaluation. We are committed to ensuring that our study is thorough and up-to-date in its comparison with relevant state-of-the-art methods.

---

### Official Review · Reviewer_1niQ · 2023-10-28

**Soundness:** 3 good
**Presentation:** 3 good
**Contribution:** 3 good
**Rating:** 5
**Confidence:** 5

**Summary:**

This paper exploits the multimodal graph representations to generate domain-invariant pivot embeddings for domain generalization. They use the graph to represent text descriptions and employ the graph matching scheme to align the embeddings of images and text. Some experiments validate the performance of the proposed method by comparing it to others.

**Strengths:**

The authors build the domain generalization problem on image datasets with a graph neural network. The overall structure of this paper includes a text graph encoder, an image graph encoder, and clustering-based graph alignment. Experimental results and visualizations on CUB-DG and DomainBed show the effectiveness of the proposed method.

**Weaknesses:**

The visual and text graph construction and the clustering-based alignment have been well and intensively studied in the current cross-modal learning, although the authors claim this is the first work for domain generalization problems.

The authors claim they suggest a novel graph neural network, but the overall graph learning is simply built on the conventional GCN or simple modification.

To me, this paper is more like a cross-modal learning or multi-modal learning algorithm rather than a domain generalization approach. The whole learning scheme mainly focuses on single model graph construction and local and global graph feature alignment.

If I vote for a cross-modal learning task, I'd like to offer a borderline acceptance, but for a new task like domain generalization, the reviewer does not see its potential for new practical tasks.

**Questions:**

What are the differences between the proposed cross-modal or multi-modal feature alignment? What are the relations between the proposed method to your final task? How could we conduct the performance evaluation?

For the global graph alignment, this is a simple metric or numerical alignment? How could we take the adversarial or another distribution alignment into account? How about the performance or novelty?

For the local alignment, we can simply take it as a cluster-level or category-level alignment? Actually, the effects will greatly overlap with the global ones. How could you figure out the local and global ones? How could we avoid the redundancy in model construction?

There are too many hyper-parameters, which are hard to deploy in practical systems, let alone special domain generalization problems.

---

> ### Author Response · Authors · 2023-11-14
> **Reply to your valuable comments [1/2]**
>
> We extend our sincere appreciation for your thorough review of ourpaper. I am truly grateful for the time and effort you dedicated to providing constructive feedback.
>
> > **Q1. What are the differences between the proposed cross-modal or multi-modal feature alignment? What are the relations between the proposed method to your final task? How could we conduct the performance evaluation?**
>
> **A1.** While our work might be perceived as centered around multimodal learning, it is crucial to underscore that our primary objective is the acquisition of domain-invariant visual features through the incorporation of domain-invariant text for domain generalization. The ultimate aim is to significantly enhance domain generalization performance, as evidenced by the results presented in Table 1 and Table 5.
>  It is noteworthy that our methodology extends beyond the realm of global alignment, encompassing the strategic incorporation of local alignment. The utilization of both global and local alignment is helpful in the challenging few-shot setting, as detailed in Table 2. By framing our work within the context of multimodal learning, we strategically leverage diverse modalities to achieve domain-invariant representations.
>  In Domain Generalization, performance evaluation proceeds as follows:
> - Select one domain as the target domain, and the remaining domains except the target domain become the source domain.
> - The model is trained using only the training data of the source domain, and the model (checkpoint)  is selected using the validation data of the source domain.
> - Use the selected model to evaluate the performance (accuracy) on the test data of the target domain. In other words, except when evaluating the model's performance, the model does not have access to any information in the target domain.
> - The final performance is the result of averaging the results when each domain is the target domain.
>
> > **Q2. For the global graph alignment, this is a simple metric or numerical alignment? How could we take the adversarial or another distribution alignment into account? How about the performance or novelty?**
>
> **A2.** In our approach, we avoid utilizing specific characteristics from each domain. Instead, we posit the existence of a feature shared across all domains. We believe that text inherently encapsulates some of this shared information. Consequently, we align the text feature with the identified pivot for enhanced model performance.

---

> ### Author Response · Authors · 2023-11-14
> **Reply to your valuable comments [2/2]**
>
> > **Q3. For the local alignment, we can simply take it as a cluster-level or category-level alignment? Actually, the effects will greatly overlap with the global ones. How could you figure out the local and global ones? How could we avoid the redundancy in model construction?**
>
> **A3.** Local alignment, in the context described, typically involves aligning features at a more granular level than global alignment. It doesn't necessarily equate to a specific category-level alignment, but rather focuses on smaller segments (clusters) within an image or text. It's about capturing fine-grained details or local patterns that might be missed by global alignment alone.
>  While global alignment considers one image and one text as singular features, local alignment takes a more granular approach. It involves dividing the image into multiple grids, treating each grid as a distinct feature, and similarly segmenting the text into multiple words, treating each word as an individual feature. Subsequently, a visual graph and a textual graph are created from these local features.
>
>  This dual alignment strategy enables our model to capture local information that might be overlooked in a purely global alignment framework. By considering both the broader context and finer local details, we enhance the alignment between images and text, consequently improving performance in domain generalization.
>  The attached figure **[(image link)](https://drive.google.com/file/d/11l39-Bv4sPmT8UI1aufzyvnbd1wasYPK/view?usp=drive_link)** provides GradCAM visualization for both GVRT and our model. It is evident that our model, incorporating graphs for local alignment, captures more diverse attributes compared to GVRT, which relies solely on global alignment without utilizing graphs.
>
> > **Q4. There are too many hyper-parameters, which are hard to deploy in practical systems, let alone special domain generalization problems.**
>
> **A4.** Indeed, as you pointed out, the paper provides many hyperparameters requiring adjustments in our model. It was my intention to convey to the readers that these parameters are subject to their control. However, among the hyperparameters outlined in the paper, we set fixed values for lambda (both $\lambda_{global}$ and $\lambda_{local}$ were set to 1, and $\lambda_{p}$, $\lambda_{h}$, $\lambda_{d}$, $\lambda_{aux}$ were set to 0.1, 0.1, 1, 1, respectively). Consequently, the practical hyperparameters involve determining the number of neighbor nodes ($K_v$, $K_t$) during the construction of visual and textual graphs, as well as deciding the number of clusters ($N_v$, $N_t$) in the graph node clustering process following graph creation.
>  In addition, although there are many hyperparameters, results on DomainBed (Table 5) show that the performance is averaged after randomly turning it into a random hyperparameter through three seeds, so it has robustness that does not fall below a certain level of performance.

---

### Official Review · Reviewer_LvEJ · 2023-10-30

**Soundness:** 3 good
**Presentation:** 3 good
**Contribution:** 2 fair
**Rating:** 5
**Confidence:** 4

**Summary:**

This paper proposes a graph-based domain generalization method to encode domain-invariant visual representations. It leverages text information and aligns the image and text from the global and local levels. Experiments on CUB-DG and PACS datasets show the effectiveness of the proposed approach.

**Strengths:**

1. This paper is well-organized and well-written.

**Weaknesses:**

1. The novelty of the proposed approach is limited. Utilizing graph models to represent the visual and text information and perform graph matching is normal in image-text retrieval task. It is unclear what is the novelty.
2. The proposed method introduces a novel modality(text information) to align different domains, which is unfair compared with those methods that only use image information. Moreover, it does not consider the relations among different domains.
3. The proposed method is unclear. Does it use the same backbone to extract features of all domains? The formulation only reflects the alignment of image and text modality, without showing different domains.
4. The authors do not compare with the art method, since the latest method in Table 1 is in 2022.
5. There is little improvement in the performance. In Table 5, the performance of using image and text information is similar to that of only using image information. Moreover, why the compared methods on the two datasets are different in Table 1? The improvements in PACS are also limited.

**Questions:**

see the weakness

---

> ### Author Response · Authors · 2023-11-14
> **Reply to your valuable comments [1/2]**
>
> We would like to express my gratitude for your valuable feedback and We appreciate the time and effort you dedicated to reviewing our paper.
>
> > **W1. The novelty of the proposed approach is limited. Utilizing graph models to represent the visual and text information and perform graph matching is normal in image-text retrieval task. It is unclear what is the novelty.**
>
> **A1.** Thank you for pointing it out. While utilizing graph models to represent visual and textual information and performing graph matching is a common approach in image-text retrieval, to the best of my knowledge, this research is the first to apply these techniques specifically in the domain generalization task. Our method leverages graph models to represent both visual and textual information, achieving improved performance in Domain Generalization (DG) tasks. Specifically, by using image-text information from the source (seen) domain, our model demonstrates strong performance even in target (unseen) domains with entirely different data distributions. This is substantiated by our outstanding results on the CUB-DG dataset (Upper table in Table 1) and the Domainbed benchmark (Table 5), where we outperform state-of-the-art models. Additionally, our approach excels in few-shot settings compared to other existing DG techniques, showcasing the robustness of our visual-textual representation based on graph structures.
>
> >  **W2. The proposed method introduces a novel modality(text information) to align different domains, which is unfair compared with those methods that only use image information. Moreover, it does not consider the relations among different domains.**
>
>  **A2.** As you rightly pointed out, comparing our model, which utilizes both text and image modalities, with models relying solely on images might be perceived as unfair. In response to this concern, we also compared our model with GVRT [1], a model that utilizes both text and image information. In the CUB-DG dataset, our model demonstrated a 2.1% higher accuracy than GVRT, with 59.1% compared to GVRT's 57% (as shown in Table 1's upper table). Similarly, on the Domainbed benchmark, our model outperformed GVRT, achieving 65.9% accuracy compared to GVRT's 65.2%, representing a 0.7% performance gain.
>
>  Furthermore, our approach aims not to analyze the characteristics of each domain individually but argues that there exists class-variant and domain-invariant information shared across all domains. We argue that the essence of domain generalization lies in uncovering domain-invariant features shared by all domains, rather than focusing solely on the correlations between individual domains. To achieve this, we leveraged text information in our model.
>
> > **W3. The proposed method is unclear. Does it use the same backbone to extract features of all domains? The formulation only reflects the alignment of image and text modality, without showing different domains.**
>
>  **A3.** We appreciate your understanding regarding any confusion stemming from the lack of clarity in our proposed method. As mentioned in Section 3.1, in the setting we studeid, we employ ResNet50 as the backbone to extract global image features and local image features across all domains, consistent with other models. This common backbone facilitates the extraction of fundamental image features in a uniform manner across different domains. If there are additional questions or if further clarification is needed on this aspect, please feel free to ask.
>
> > **W4. The authors do not compare with the art method, since the latest method in Table 1 is in 2022.**
>
> **A4.** As far as we are aware, MIRO [2] is the most recent SOTA paper in this setting. To ensure a comprehensive comparison, we have included experimental results that encompass MIRO. However, if there are other recent methods that you suggest we consider for comparison, please provide the references, and we will gladly incorporate them into our evaluation. We are committed to ensuring that our study is thorough and up-to-date in its comparison with relevant state-of-the-art methods.
>
> [1] Seonwoo Min, Nokyung Park, Siwon Kim, Seunghyun Park, and Jinkyu Kim. Grounding visual
> representations with texts for domain generalization. In Computer Vision–ECCV 2022: 17th
> European Conference, Tel Aviv, Israel, October 23–27, 2022, Proceedings, Part XXXVII, pp. 37–
> 53. Springer, 2022.
> [2] Junbum Cha, Kyungjae Lee, Sungrae Park, and Sanghyuk Chun. Domain generalization by mutualinformation regularization with pre-trained models. In Computer Vision–ECCV 2022: 17th European Conference, Tel Aviv, Israel, October 23–27, 2022, Proceedings, Part XXIII, pp. 440–457.
> Springer, 2022.

---

> ### Author Response · Authors · 2023-11-14
> **Reply to your valuable comments [2/2]**
>
> > **W5. There is little improvement in the performance. In Table 5, the performance of using image and text information is similar to that of only using image information. Moreover, why the compared methods on the two datasets are different in Table 1? The improvements in PACS are also limited.**
>
> **A5.** As you pointed out, it may initially appear that the improvement on DomainBed is not substantial. In the CUB-DG dataset, our model (SOTA) surpasses the ERM [1] by 13.6%. In the context of DomainBed, the performance disparity between our model (SOTA) and ERM is 2.6%. It's worth noting that in DomainBed, the absolute performance change may not be as pronounced. For example, Table 1 shows the top five methods in DomainBed's PACS dataset. It is meaningful that the performance difference between our model and the existing sota is 0.7%, compared to the 0.1% performance difference between the existing sota, SagNet [2], and the previous sota, mDSDI [3]. (ERM is indeed the most basic and straightforward approach, utilizing the average classification loss across each source (seen) domain as the overall loss for training. In essence, it represents the general classification.) It is noteworthy that our model attains the SOTA in both CUB-DG and DomainBed, showcasing its effectiveness across various settings. Moreover, our model consistently outperforms other methods, even in challenging few-shot scenarios. Our model outperformed other methods in the few-shot setting. (Table 2)
>
>  We also understand your point regarding the inconsistency in the compared methods across different datasets in Table 1. This discrepancy arises from paper limitations, leading us to present the top five performing methods for brevity. For a more comprehensive view, the complete versions of Table 1 (upper and lower), and Table 5 can be found in the appendix (Table 7, Table 10, Table 8).
>
> [1] Vladimir N Vapnik. An overview of statistical learning theory. IEEE transactions on neural networks, 10(5):988–999, 1999
> [2] Hyeonseob Nam et al. Reducing domain gap by reducing style bias. In CVPR, 2021
> [3] Manh-Ha Bui, Toan Tran, Anh Tran, and Dinh Phung. Exploiting domain-specific features to enhance domain generalization. Advances in Neural Information Processing Systems, 34:21189–
> 21201, 2021.

---

### Author Response · Authors · 2023-11-14
**To all reviewers**

Dear Reviewers,

We extend our sincere appreciation for the time and effort you dedicated to reviewing our work.

Allow us to summarize the key points of our paper.

 Our research delves into the Domain Generalization task, addressing the challenge of empowering a model to perform effectively in a target domain (unseen domain) with a novel data distribution, despite its initial training on data from a source domain (seen domain). This challenge becomes particularly critical when disparities exist between the distributions of the training and test data. Our approach centers on domain generalization in image classification, leveraging domain-invariant text to compensate for domain-variant images.

By using a text feature as a pivotal point, we implemented a comprehensive alignment strategy, encompassing both global and local alignment, to align visual representations with textual. To enrich the model's understanding of diverse attributes, we constructed visual and textual graphs from local representations and integrated the alignment of local features between text and image by matching the visual graph and textual graph.

Our methodology yielded state-of-the-art (SOTA) performance on the CUB-DG dataset and the large-scaled benchmark, DomainBed. Notably, in contrast to comparable methods, our approach maintains robust performance even in a few-shot setting, underscoring its versatility and effectiveness across diverse scenarios.

Once again, we sincerely appreciate your thoughtful evaluation, and we welcome any feedback or inquiries you may have. Your insights significantly contribute to refining and advancing our research.